



# 1 Seiche excitation in a highly stratified fjord of southern

# 2 Chile: the Reloncaví fjord.

Manuel I. Castillo[1*], Oscar Pizarro[2,3,4] Nadin Ramírez[2,4] and Mario Cáceres[1]
[1]{Facultad de Ciencias del Mar y Recursos Naturales, Universidad de Valparaíso,
Valparaíso, Chile.}.
[2] {Programa COPAS-Sur Austral, Universidad de Concepción, Concepción, Chile.}
[3] {Departamento de Geofísica, Universidad de Concepción, Concepción, Chile.}
[4] {Instituto Milenio de Oceanografía, Universidad de Concepción, Chile.}

19 *Correspondence to: manuel.castillo@uv.cl



**Abstract**
We describe a seiche process in a Chilean fjord, based on current, temperature and sealevel
data obtained from the Reloncavi fjord (41.6° S, 72.5° W) in southern Chile. We combined
four months of ADCP data with sealevel, temperature and wind time series to analyze the
dynamics of low-frequency (periods > 1 day) internal oscillations in the fjord. Additionally,
seasonal CTD data from 19 along-fjord stations were used to characterize the seasonality of
the density field. The density profiles were used to estimate the internal long-wave phase
speed ($c$) using two approximations: (1) a simple reduced gravity model (RGM) and (2) a
continuously stratified model (CSM). No major seasonal changes in $c$ were observed using
either approximation (e.g., the CSM yielded $0.73 < c < 0.87$ m s$^{-1}$ for mode 1). The natural
internal periods ($T_N$) were estimated using Merians's formula for a simple fjord-like basin
and the above phase speeds. Estimated values of $T_N$ varied between 2.9 and 3.5 days and
were highly consistent with spectral peaks observed in the along-fjord currents and
temperature time series. We conclude that these oscillations were forced by the wind stress,
despite the moderate wind energy. Wind conditions at the end of winter gave us an excellent
opportunity to explore the damping process. The observed damping time ($T_d$) was relatively
long ($T_d = 9.1$ days).
**1    Introduction**
Internal seiche oscillation has long been known in closed basin geometries (e.g. Watson
1904; Wedderburn 1907; Wedderburn and Young 1915). The first detailed description
thereof was presented by Mortimer (1952). In these systems, wind is the main force affecting
the surface and isotherms (Wiegand and Chamberlain, 1987), which produces a set of
periodic oscillations and circulation cells throughout the water column that may contribute to
internal mixing of the basin (Thorpe 1974; Monismith 1985; Wiegand and Chamberlain
1987; Munnich et al. 1992; Mans et al. 2011; Simpson et al. 2011).
Although external (barotropic) seiches are ubiquitous in closed basin geometries (Münnich et
al. 1992), it is not theoretically evident that there are internal seiches (baroclinic) in a linearly
stratified fluid (Maas and Lam 1995). It is possible to find resonant basin modes only in



well-behaved geometries (Arneborg and Liljebladh, 2001a).. However, studies of lakes have
yielded good results using layered models (e.g. Lemmin 1987), normal-mode approximations
(e.g. Wiegand and Chamberlain 1987; Münnich et al. 1992) or numerical model simulations
(e.g. Goudsmit et al. 2002). In fact, internal seiches have been observed in semi-enclosed
systems such as fjords (e.g. Djurfeldt 1987; Pasmar and Stigebrandt 1997; Arneborg and
Liljebladh, 2001a) with complex geometries and where linear stratification is rarely
observed, and thus the only way to maintain consistency with the theory is that the
oscillation on the pycnocline dominates the internal seiche oscillation (Arneborg and
Liljebladh, 2001a). Early in the development of a seiche, its amplitude is related to the
forcing intensity, and the standing oscillation then becomes free and requires no additional
forcing. The frequencies are retained, but the amplitudes decays (damping) exponentially due
to friction until the system comes to rest (Rabinovich 2010). The development of seiche
oscillations depends of the forcing and damping mechanisms; with large damping, it is
impossible to observe a seiche, whereas small damping of a seiche allows for several
oscillations (Arneborg and Liljebladh, 2001a).
In fjords with shallow sills, the interaction between the sill and the barotropic tide generates
internal tides that are more energetic than other internal oscillations and are the focus of most
studies regarding mixing and internal oscillations based on internal tides (e.g. Stigebrandt
1980; Stigebrandt and Aure 1989; Inall and Rippeth 2002; Ross et al. 2014). In the case of
fjords with a deep sill and low tidal energy, the breaking of the internal seiche oscillations at
the boundaries could be an important contributor to the internal mixing, promoting the
spreading of properties within the fjord, particularly in deep waters (Stigebrandt and Aure
1989; Münnich et al. 1992; Arneborg and Liljebladh 2001b). Additionally, there are
evidences that vertical isopycnal displacements in fjords could be generated by similar
displacements outside the fjord (e.g. Svensen 1980; Djurfeldt 1987). These remotely
generated oscillations could enhance the mixing and ventilation in deep fjords.
There is still only limited understanding of the main oceanographic processes occurring in
the fjord region of southern Chile, although there has been local research during the previous
few decades. Since early studies of the hydrography by Pickard (1971), a systematic



measurement program in the fjord region has been maintained since 1995 (Palma and Silva
2008; Pantoja et al. 2011; Iriarte et al. 2014), although only a small number of studies have
focused on the physical dynamics. Most studies have been conducted over short time spans
(e.g. Cáceres et al. 2004; Valle-Levinson et al. 2007), and only a few studies have been based
on more than one month of data (e.g. Valle-Levinson and Blanco 2007; Letelier et al. 2011;
Castillo et al. 2012; Schneider et al. 2014), thereby limiting our understanding of sub-inertial
variability. In the Reloncavi fjord, time series of approximately 4 months have shown
evidence that 3-day oscillations of currents could be produced by internal seiche oscillations
(Castillo et al., 2012) but lack on the forcing to describe the forcing mechanism and the
seasonal modulation.
This study presents the first evidence of internal seiche oscillations in a fjord in southern
Chile. The objective of this study was to address the following questions: How do these
oscillations affect the temporal and spatial dynamics of currents and temperature? How are
these oscillations forced?
**2    Study area**
The Reloncavi fjord (41.5ºS, 72.5ºW) is the northern most fjords on the coast of Chile (Fig.
1). This "J" shaped fjord is 55 km long and has a width that varies from 3 km near the mouth
to 1 km near the head. There is a deep sill (~ 200 m depth) located 15 km inland although it
does not appear to be a barrier to the exchange of properties between the adjacent basins.
Based on bathymetric features and the coastline morphology, this fjord can be separated into
four sub-basins displaying the characteristics presented in Table 1 and figure 2.
The main river discharge is provided by the Puelo River (at the middle of the fjord), which
produces a mean annual discharge of 650 $m^3s^{-1}$.The Petrohue River (at the head of the fjord)
has an mean annual discharge of 255 $m^3s^{-1}$, and there are additional freshwater inputs of
minor importance compared with the Cochamo river (mean annual discharge of 20 $m^3s^{-1}$)
and Canutillar hydroelectrical plant (mean annual discharge 75.5 $m^3s^{-1}$) (Niemeyer and
Cereceda, 1984). The freshwater input to the fjord due to direct precipitation is only





approximately 2% of the main river discharge (León-Muñoz, 2013), and its contribution may
be in balance with evaporation (Castillo et al., 2016). The freshwater input creates a marked
along-fjord pycnocline that is deeper at the head (~8 m) and shallower at the mouth (~3 m)
(Fig. 2).
During the winter, the mean wind stress ($\tau$) is low due to calms winds ($< 10^{-3}$ N m$^{-2}$). During
storm events in winter, $\tau$ can reach values as high as 0.4 N m$^{-2}$ (winds of $> 10$ m s$^{-1}$), and the
wind tends to blow out of the fjord, thereby reinforcing the upper outflow of brackish water.
In contrast, during the spring/summer, the winds exhibit a marked diurnal cycle, and $\tau$ can
reach values as high as those observed in the winter, whereas the wind blows landward, i.e.,
toward the fjord's head and against the upper flow. Tides in the Reloncavi fjord are
predominantly semi-diurnal, and during spring tidal range never exceed 6 m, whereas the
neap tidal range is about 2 m. The tidal current is relatively weak in the upper layer, which is
dominated by gravitational circulation (Valle-Levinson et al., 2007; Montero et al., 2011;
Castillo et al. 2012).
**3      Data and Methods**
**3.1      Field Observations**
Current measurements were obtained using Teledyne RD Instruments Acoustic Doppler
Currentmeter Profilers (ADCPs) in three subsurface mooring systems. These subsurface
systems were located near the fjord mouth, near the Puelo River and between the Cochamo
and Petrohue Rivers (Fig. 1). The longest time series spanned the period of August through
November 2008 (Fig. 1 and Table 1). At the mouth, two upper-looking ADCPs were
positioned at nominal depths of 10 m (300 kHz) and 450 m (75 kHz). The Puelo mooring
held two ADCPs, one facing-up at a depth of 30 m (600 kHz) and one facing downward at a
depth of 35 m (300 kHz). The Cochamo mooring held one facing-up ADCP at a depth of 11
m (300 kHz). Note that due to the large tidal range, the depths of the ADCPs significantly
changed with the tides. These effects — along with small vertical deviations of the ADCPs
related to the line movements — were corrected using the ADCPs pressure sensors, and all
of the bin depths were referenced to the water surface level. The mooring systems were



designed to obtain the best vertical resolution available with emphasis on the upper layer.
The ADCP cell sizes were 0.5 m (600 kHz), 1 m (300 kHz) and 4 m (75 kHz), and the data-
acquisition time intervals were 10 minutes in most of the ADCPs, with the exception of the
deepest ADCP, which was set to acquire data at an interval of 20 minutes. All the ADCPs
configurations maintain a standard deviation$< 2$ cm s$^{-1}$ (details in supporting information
S2).
The morphology of the fjord exhibits a sharp bend in the middle, and thus the $x$ and $y$-
components of the currents were rotated to the local orientation of the along-fjord axis (Fig.
1 and Table 1). A right-handed coordinate system with a positive-up $z$-axis and an along-
fjord $y$-axis (positive toward the fjord head) was used. Consequently, the cross-fjord $x$-
component was positive toward the south (east) near the fjord mouth (head). To assess the
contribution of the tides to the currents, the amplitudes and phases of several tidal
components were calculated at all of the moored ADCPs using a standard harmonic analysis
from Pawlowicz et al. (2002).
The vertical structure of the temperature was obtained from Onset HOBO-U22 temperature
sensors installed in three mooring systems along the fjord (Fig. 1). These moorings held
surface buoys supporting the thermistor chains with an anchor located at a 25 m depth to
maintain their nominal depths (0, 1, 2, 3, 4, 5, 7, 9, 11, 13, 15 and 20 m) from the surface
independent of tidal fluctuations. Temperature data were collected every 10 minutes at all
locations.
A Davis Vantage Pro2 meteorological station was installed south of the Puelo River (see Fig.
1). This station held sensors for measuring the wind direction and velocity, solar radiation,
rain, and air temperature. The wind magnitude and direction sensors were installed 10 m
above sealevel and were set to collect data every 10 minutes from 12 June 2008 to 30 March
2011. Gaps in the time series represented only 0.04% of the total data. The wind stress ($\tau$)
was calculated using a drag coefficient dependent on the magnitude (see Large and Pond,
1981) and a constant air density of 1.2 kgm$^{-3}$.



The salinity and temperature profiles were obtained seasonally using a CTD SeaBird SBE 25
at 19 stations in the along-fjord transect shown on Figure 1. The data were processed
following the standard protocol suggested by the manufacturer and were averaged in vertical
intervals of 0.5 m. Due to large salinity changes in the upper layer, the instrument pump was
set to a time interval of 1 minute. After the start of the pumping, the instrument was
maintained near the surface until the sensors stabilization. Then, the CTD was lowered to the
maximum depth of the station (Table 2). The along-fjord transects typically required 12 to 24
hours to complete, depending on local weather conditions. Due to technical limitations, the
winter transect was performed to a maximum depth of 50 m.
The sealevel was recorded every 10 minutes using two pressure sensors moored over the
seabed. At Cochamo, the pressure sensor was an Onset HOBO-U20, whereas a SeaBird
wave-tide gauge SBE-26 was installed near the fjord's mouth (Fig. 1). Subsurface pressure
data were corrected for air pressure and converted to an adjusted sealevel.
Discharge data were provided by Dirección General de Aguas, Chile (Dirección General de
Aguas, 2016). These data are regularly collected at a station located 12 km upstream of the
Puelo River's mouth (Fig. 1). The time series extended from January 2003 to December
2011, and data gaps represented only 2% of the total.
**3.2    Time series analysis**
Previous findings (Castillo et al., 2012) have shown an important oscillation with a period of
approximately 3 days (72 h).To focus the study on these perturbations, we used a cosine-
Lanczos band-pass filter with half amplitudes at 60 h and 100 h (see results for the
justification of the selected band). As part of the results, the band-passed time series of the
current and temperature data are shown (COPAS-SUR Austral, 2012).
Spectral analyses of the current, wind stress, sealevel and temperature time series were
performed using Welch's modified average periodograms (Emery and Thomson 1998). To
achieve statistical reliability of the spectral estimations, each time series was divided into
non-overlapping segments to generate spectral estimates. In the case of the current time



series, the spectra were (additionally) averaged among depth layers to obtain 12, 24 and 48
degrees of freedom, depending on the frequency (see Fig. 3). In addition, to evaluate the
consistency of the periodicity between the time series, we calculate a Morlet cross-wavelet
analysis following wavelet methods explained by (Torrence and Compo, 1998) and (Grinsted
et al., 2004).
The phase velocity ($c$) was estimated using two models that took into account the fjord
stratification: (1) a simple reduced-gravity model (RGM) and (2) a continuously stratified
model (CSM).
The reduced-gravity model was developed using the typical density profiles in each sub-
basin. Here, the base of the upper layer was estimated from the pycnocline depth (Fig. 2),
which in the Reloncavi fjord is well represented by the depth of the 24 isohaline ($h_1$)
(Castillo et al., 2016), considering that $h_1$ is the pycnocline depth and $H$ is the deepest CTD
cast (mostly near to the sub-basins maximum depths). The mean density of the upper layer (
$\rho_1$ ) was estimated from depths between the surface to $h_1$, whereas the mean density for the
deep layer ( $\rho_2$ ) was estimated for depths between $h_1$ and $H$. These estimations were made
for each sub-basins, and seasons (Table 2).
Using both densities, $\rho_1$ and $\rho_2$, the reduced gravity ( $g' = g(\rho_2 - \rho_1)/\rho_2$ ) was obtained,
here $g$ is the acceleration of gravity. The internal phase velocity of each sub-basin,
$c_i = \left( g' \ h_{1i} \right)^{1/2}$, where $i$ =1 to 4 and $h_{1i}$ represents the mean depth of the upper layer in the
sub-basin "$i$" was used to estimate a mean wave speed in the entire fjord,

$$T = \frac{L}{c} = \sum_{i=1}^{n} \frac{L_i}{c_i} \tag{1}$$

where $L_i$ is the $i$ sub-basin length and $L$ is the fjord length.
The continuously stratified model (CSM) was developed using the normal mode analysis,
which introduced the stratification as $N^2 = -(g/\rho)(\partial \rho / \partial z)$, which is the buoyancy
frequency, in the Sturm-Liuoville expression



$$\frac{d}{dz}\left(\frac{1}{N^2}\frac{d\psi_n}{dz}\right)+\frac{1}{c_n^2}\psi_n = 0 \tag{2}$$

where $\psi_n(z)$ is the vertical structure of the horizontal velocity for the mode $n$. Here $c_n$
represents the $n$ mode speed (see Gill, 1982) and differs significantly from phase speed if
rotation plays a role (van der Lee and Umlauf, 2011).
Independent of the model used to obtain the phase speed (RGM or CSM), the natural
oscillation period ($T_N$) was determined using Merian's formula for a semi-enclosed basin, as
suggested by Ravinovich (2010), $T_N = 4\ T$.
The modal decomposition was used to obtain the contribution of each mode in the currents
variability (e.g. Emery and Thomson, 1998; Gill, 1982; van der Lee and Umlauf, 2011). The
along- and cross-fjord band-pass currents [$u_{bp}$, $v_{bp}$] could be described by the vertical modes
by (3),

$$[u_{bp},v_{bp}](z,t) = \sum_{n=1}^{\infty}[u_{pj},v_{pj}](t)\ \psi_n(z) \tag{3}$$

The along- and cross-fjord currents projected ($u_{pj}$, $v_{bp}$) on the vertical modal structure ($\psi_n$) was obtained by eq. (4),

$$[u_{pj},v_{pj}](t) = \frac{1}{H}\int_{-H}^{0}[u_{bp},v_{bp}](z,t)\ \psi_n(z)\ dz \tag{4}$$





## 4. Results

### 4.1 Density structure

As a result of abundant freshwater input to the fjord, there were marked differences in density between the upper and lower layers along the fjord and small changes in stratification among seasons, particularly near the mouth of the fjord (Fig. 2). One important characteristic of the upper layer is its high and persistent stratification from the surface to the base of the pycnocline (Fig. 2). Along the fjord, the pycnocline depth exhibited clear deepening from 2.3 ± 0.1 m at the mouth to 6.1 ± 0.3 m near the head. The pycnocline depth exhibited greater seasonal variability near the head of the fjord (Fig. 2).

### 4.2 Winds, sealevel and freshwater discharge

The along-fjord wind stress ($\tau$) displayed two patterns during the transition from winter to spring. During the winter, $\tau$ was generally out of the fjord (-0.4 ± 3 x$10^{-2}$ N m$^{-2}$) and displayed oscillations with a period longer than 1 day. There were also strong events (> 0.2 N m$^{-2}$) during the first half of August 2008 that could be associated with the end of winter storms in the region. This winter pattern drastically changed during the early spring (first week of September 2008) and was maintained throughout the rest of the season. Changes were evident in a marked daily cycle and in switches from down- to up-fjord (average of 1.6 ± 3 x$10^{-2}$ N m$^{-2}$), against the upper layer outflow (Fig. 3a).

The sealevel was measured at the mouth and near Cochamo (Fig. 1). At both stations, the form factor was 0.12, which indicates that semi-diurnal tides dominate in the region. In fact, the $M_2$ amplitude was 1.89 ± 0.06 m at the mouth and 1.91 ± 0.06 m near Cochamo. The mouth-to-head phase difference in this harmonic was negative (-2.4º), indicating propagation toward the head with a lag of approximately 5 minutes. The maximum tidal range during spring tides was approximately 6 m and less than 1 m during neap tides (Fig. 3b). Similar ranges have been observed outside the fjord in the Reloncavi sound (Aiken, 2008).





Discharge was greatest (approximately 1413 m$^3$ s$^{-1}$) at the end of August 2008 (winter) and
lowers (approximately 459 m$^3$ s$^{-1}$) at the end of October (spring). In the winter, the historical
mean of 650 m$^3$ s$^{-1}$ (Niemeyer and Cereceda 1984; Leon et al 2013) was exceeded 86% of
the time, whereas during the spring, this exceedance occurred only 18% of the time. In fact,
only a small variability around the mean was observed during the spring (Fig. 3c).
**4.3   Along-fjord currents**
The along-fjord currents were one order of magnitude larger than the cross-fjord currents (in
this study we focused on the along-fjord component). At the three measurements sites at
Cochamo (Fig. 3d), Puelo (Fig. 3e) and the mouth (Fig. 3f), the along-fjord currents
displayed certain common features: (1) semi-diurnal oscillations attributed to tidal effect, (2)
a two layered structure with persistent outflow above the pycnocline and an intermittent
lower inflow layer beneath, and (3) several low-frequency (period > 1 day) oscillations were
present in the time series.
Currents in the upper outflow layer displayed a mean velocity of 66 cm s$^{-1}$ at the mouth and
45 cm s$^{-1}$ at Cochamo, indicating that the outflow increased through the mouth. Additionally,
the upper layer was deeper at Cochamo (Fig. 3d) than at the mouth (Fig. 3f), which is
consistent with the along-fjord pycnocline depth (Fig. 2). Below the upper layer, a sub-
surface layer displayed intermittent inflow (see Fig. 3d, 3e and 3f) with a maximum (> 20 cm
s$^{-1}$) centered at the ~ 6 m depth.
This two-layered pattern was clearly observed in the upper 10-15 m and is consistent with a
gravitational circulation due to the along-fjord pressure gradient. This pressure gradient is
also consistent with the observed along-fjord pycnocline tilt (Fig. 2). At depths > 20 m, the
along-fjord currents at Puelo and at the mouth exhibited an important influence (> 40% of
the variability) of a semi-diurnal component of the tide. In addition, in this layer, low-
frequency (periods > 7 days) oscillations suggest a bottom-to-surface propagation that was
more intense from the end of August to the beginning of September during a period of high





discharge ($> 650$ m$^3$ s$^{-1}$). This layer on average exhibited a weak outflow ($\sim 1$ cm s$^{-1}$) at the
mouth, which in turn implies a 3-layer pattern of the residual flow near the mouth.
**4.4     Spectral characteristics of currents, temperature, sealevel and winds**
To obtain better statistic reliability, the spectra of the along-fjord currents were depth-
averaged. The upper layer was defined until the pycnocline depth ($z \leq h_1$), whereas the deep
layer contains $z > h_1$ (Fig. 4).
All of the spectra displayed an energetic peak at the semi-diurnal frequency ($M_2$), and this
peak was greater in the deep layer (Fig. 4). In the diurnal band, the spectra at Puelo and at the
mouth presented a clear (and highly energetic) peak in the surface layers. This diurnal peak
is likely due to the influence of wind stress (see Fig. S1), which displayed a marked diurnal
cycle during the late winter (end of August) and spring (Fig. 3a). An important peak ($10^4$ cm$^2$
s$^{-2}$ cph$^{-1}$) was observed only at Cochamo in the 6 hour band ($M_4$), suggesting an increase in
the importance of non-linear interaction between $M_2$ and the bathymetry in this sub-basin.
The spectra in the upper layer displayed an important accumulation of energy in the band
centered on the 3days period. The band was wider (between 2 and 7 days) at the mouth and
Puelo and narrower (between 1.5 and 4 days) at Cochamo. At the mouth, the maximum
spectral density was in the 3 days band ($> 10^5$ cm$^2$ s$^{-2}$ cph$^{-1}$) and was one order greater than
the maximum spectral density observed at Cochamo ($\sim 10^4$ cm$^2$ s$^{-2}$ cph$^{-1}$). Another important
accumulation of energy in the along-fjord currents was centered on the 15 days period. One
characteristic of the 15 days band is the influence on the entire water column at Puelo and the
mouth (Fig. 4).
The sealevels at Cochamo ($\eta_C$) and at the mouth ($\eta_m$) were similar at frequencies less than
0.165 cph (periods longer than 6 h). The spectra displayed an important accumulation in the
synoptic band (10 days). Both locations exhibited the same energy at the diurnal ($K_1$)
semidiurnal ($M_2$) frequencies, although $M_2$ was clearly the dominant harmonic in the fjord.
The spectral energy was one order of magnitude higher than the diurnal ($K_1$) harmonics and
three orders of magnitude higher than the quarter-diurnal ($M_4$) harmonics. The spectra



exhibited no accumulation of energy in the 3days band, although at high frequencies (> 0.5
cph), an important accumulation of energy was observed in the 1.3h band (between 1.16 h
and 1.56 h) at $\eta_C$ (Fig. 4).
The wind stress ($\tau$) indicated that the along-fjord wind stress was significantly higher than
the cross-fjord component. The spectra displayed a marked peak (particularly in the along-
fjord component) in the diurnal band, which is likely due to the sea-breeze phenomenon.
Another interesting feature of the spectrum was the peak in the semi-diurnal frequency,
which was observed in both components. At longer periods (> 1day), the along-fjord wind
stress displayed an important but not statistically significant peak at 2.8 days, which is highly
consistent with the currents (Fig. 4).
**4.5**    **Seasonality of the internal oscillations**
The density structure on the fjord does not show an upper mixing layer along the seasons;
indeed a continuously stratified upper layer is present along the seasons (Fig. 5). The along-
fjord mean of the pycnocline depth ($h_1$), which was estimated based on salinity/density
gradient, was used to estimate the internal phase velocity ($c$) and the internal period ($T_N$).
Seasonally, $h_1$ does not change significantly during winter, spring and summer (between 4.6
and 4.8 m) but was shallower during autumn (~ 4.1 m) (Table 2). In addition, the density
structure showed a condition of continuous stratification in the upper layer along the seasons
(Fig. 5).
In the case of the RGM approximation, internal phase velocities ($c$) were highest during
spring and summer (> 0.83 m s$^{-1}$) whereas in winter and autumn the intensities were < 0.76
m s$^{-1}$, thus we obtain internal periods between 2.9 and 3.4 days (70 and 82 hours) (Table 2).
The horizontal velocity structure ($\psi_n$) profile of the first 3 internal modes obtained from the
CSM, showed high consistency along the fjord (in each sub-basin) and through the seasons
(Fig 5). The mode 1 was highly baroclinic changing of sign nearly of 10 m (sub-basin I) and
15 m (sub-basin IV). In the case of mode 2 and 3, relatively high variability along the



seasons was observed specially at the sub-basins I and IV above of 20 m depth. For depths >
30 m (not shown) the internal modes do not show significant variability (Fig. 5). The modal
speeds for the first 3 modes described above were relatively high during spring and summer
($c_1$ was > 0.84 m s$^{-1}$) and lower during winter and autumn (here $c_1$ was < 0.77 m s$^{-1}$). These
results were highly consistent with the internal speeds obtained by RGM (Table 2).
As the internal speeds ($c$), the natural internal period ($T_N$) obtained by RGM with the mode 1
of CSM were highly consistent. For comparison, we take into account $T_N$ obtained from the
mode 1 of the CSM which ranged between 2.9 days (spring) and 3.5 days (winter). The
estimations of $T_N$ with RGM showed speeds between 2.9 days (spring) and 3.4 days (winter
and autumn) in hours this range was between 70 and 84 hours, indicating that oscillations
between these periods are dominated by mode 1 internal seiche oscillation.
To focus on these internal seiche oscillations, we filtered the along-fjord currents with a 70h
to 90h cosine-Lanczos band-pass filter. Additionally, mode 1 of the internal seiche was
associated with the pycnocline depth, which is restricted to the upper 8 m (Fig. 2). Therefore,
we describe the along-fjord currents in the upper 10 m (Fig. 6).
Vertical pattern at the three locations shows inflow/outflow intermittence along the whole
time series, also mostly of these along-fjord structures seems to develop an inclination which
suggest the baroclinic nature of this pattern. The band-pass along-fjord currents were intense
at the mouth (> 15 cm s$^{-1}$) but diminish toward the head. Intense perturbations oscillations
were observed near the surface between 10 and 20 August 2008 at the mouth and Cochamo,
internal intensification (between 4 m and 10 m depths) of the inflow/outflow pattern was
clear at Puelo and Cochamo at the ends of September. To confirm whether the nature of the
along-fjord currents pattern was baroclinic or barotropic we used $\psi_n(z)$ to project the band-
pass currents (eq. 3 and 4), similar to van der Lee and Umlauf (2011).
As an example, for the Cochamo along-fjord currents, the projection using only the mode 1,
represented 44% of the band-pass variability. The use of the first three internal modes
explained the 73% of the variability. Similarly, the barotropic mode only takes into account





of the 5% of the variability indicating the baroclinic nature of the band-pass pattern of the
along-fjord currents.
The projected along-fjord currents (modes 1-3) shows clearly the intensifications at the
surface of the middle of August, and the internal intensification of the ends of September,
also the currents at the mouth were more intense than Cochamo and the complex vertical
structures of the inflow/outflow were well defined by the projections (Fig. 7). In addition,
we estimated the kinetic energy ($K_E$) of the projected along-fjord currents for the first 3
baroclinic modes by,
$$K_E = \frac{1}{2}\sum_{n=1}^{m}(u_{pj}^2 + v_{pj}^2) \qquad (5)$$
The vertical averaged $K_E$ at the mouth was higher than Cochamo, here the total $K_E$ represent
only 9% of the total $K_E$ at the mouth. The quoted inflow/outflow intensification was also
observed in $K_E$. During the August intensification $K_E$ at Cochamo might be 4% of the mouth,
whereas during September $K_E$ at Cochamo might be as large as 15% to the mouth (Fig. 7).
Along-currents were highly coherent at 3 days band which is the period of the first mode of
the internal seiche (Table 2). To describe the variability of this high coherence along the
time, we selected 3 m depth ADCP bins (on the upper layer) from the mouth, Puelo and
Cochamo to make a Morlet cross-wavelet analysis and to estimate the squared coherence
(only refer as coherence hereafter) and phase spectrums for the relations mouth/Puelo (MP)
(Fig. 8b, 8c) and Puelo/Cochamo (PC) (Fig. 8d, 8e). Both relations showed high coherence in
the semi-diurnal and diurnal band especially during spring-tides.
At the 3 day band, the MP relation shows significant coherence during into the fjord winds
whereas low coherence was observed during the opposite winds (Fig. 8a and 8b). Similarly,
the coherence for the PC relation was high along the 3 days band except during the change of
the wind direction described above (Fig. 8d). The associated phase spectra (only the
significant coherence) at the 3 days band was ~ 0° indicating that the oscillation is in phase
along the fjord (Fig. 8c and 8e).



At the beginning of the time series, intense fluctuations were observed at Cochamo and at the
mouth (Fig. 6). To explore their relationship with the wind forcing, a detailed view of the
period between 8 and 31 August 2008, is presented in Fig. 9. During this period, the along-
fjord wind stress (not filtered) displayed three different states: (a) strong ($> 0.2$ N m$^{-2}$) up to
the fjord winds, (b) weak ($< 0.1$ N m$^{-2}$ ) or nearly calm winds and (c) moderate ($\sim - 0.1$ N m$^{-}$
$^{2}$) down to the fjord winds. During (c), the winds displayed an apparent diurnal cycle (e.g.,
Fig. 3a).
Although density is dominated by salinity, here we used temperature moorings to emphasize
that the internal oscillation reported here had an expression in other properties of the water
within the fjord. In addition, the band-pass temperature time series and the along-fjord
currents shows consistent oscillations pattern (Fig. 9). During (a), the upper outflows
weakened due to the opposing winds at the surface. This change reached depths consistent
with the pycnocline (Fig. 2), caused a disruption and subsequently forced the internal
oscillations observed in the currents and temperature fields (Fig. 9). Here, intense
perturbations were observed that weakened the surface outflow and introduced the colder
water of the upper layer to depths $> 2$ m at Cochamo and Puelo. During (b), the upper
outflow displayed minimum perturbations in both the currents and temperature. In (c),
perturbations in the currents and temperature were evident at Cochamo and at the mouth with
no major oscillations at Puelo (Fig. 9).
**5      Discussion**
We used data from one of the most extensive study ever conducted in a Chilean fjord. The
data included currents (ADCPs) and temperatures from moored instruments, seasonal CTD
information and times series of winds and sealevel to study the dynamics of the internal
seiche oscillations in the Reloncavi fjord.
In fjords with shallow sills such as the Gullmar fjord in Sweden (Arneborg and Liljebladh,
2001a), the Knight Inlet in Canada (Farmer and Freeland, 1983) and the Aysen fjord in Chile
(Cáceres et al., 2002), internal tide oscillations may play a key role in the internal mixing



(e.g. Stigebrandt 1976; Farmer and Smith 1980). In lakes, large internal seiche oscillations
significantly contribute to the mixing of the entire basin (Cossu and Wells, 2013), and these
oscillations could also be important in fjords where the relative importance of internal tides
may be less than the internal seiche oscillations (Arneborg and Liljebladh, 2001b).
In this study, we demonstrate the presence (and persistence) of seiches in a Chilean fjord
based on the sealevel slope (barotropic seiche), currents and temperatures (internal seiche).
We also studied the main processes forcing the natural oscillation of the pycnocline.
At high frequencies, the tidal spectrum (Fig. 5) displayed a clear accumulation of energy
centered at a period of 1.3 h. This frequency is not related to any tidal harmonic interactions
(Pawlowicz et al., 2002), and the shape of the spectrum (it is not a peak) suggests resonance
in this frequency band. We explored the effect of the natural oscillation of the basin in this
pattern using the barotropic phase velocity ($c$) for a shallow water wave $c = (gh)^{1/2}$ , where $h$
is the mean depth of the fjord. If one assumes a mean fjord depth of $h = 250$ m (Table 1),
then $c = 49.5$ m s$^{-1}$, and the natural period $T_N = 4L\ c^{-1} = 1.24$ h. This period is lower than the
observed period in Fig. 5 (1.3 h) because the mean depth takes into account the entire fjord
bottom profile (Fig. 1), and thus the effective depth (up to Cochamo) was 233 m and it is
closer to the 226 m necessary to obtain the observed period in Fig. 5. Winds in the region are
moderate (see Fig. 3), but their intensity is sufficient to tilt the surface slope at Cochamo
(Castillo et al., 2012), and thus the surface of the fjord oscillates with the natural period of
the basin. Further evidence of this pattern is provided by the clear differences in amplitude of
the sealevel spectrum at Cochamo (near the fjord's head) and at the mouth. This association
is attributed to the dynamics of seiches in fjords, which tend to produce a node at the mouth
and an anti-node at the head (Dyer, 1997). At the node, the sealevel amplitude must be zero,
whereas near the head, it must be a maximum. This pattern is highly consistent with the
observed spectra at 1.3 h (Fig. 5). Based on all of these results, we suggest that oscillations
close to 1.3 h will resonate with the natural period along the fjord.
Daily winds were highly coherent with surface along-fjord currents, especially on the
brackish water layer (S1). During the spring, daily periodicity of winds was strong (Castillo



et al., 2016) with intensities capable of perturb the pycnocline and to induce the internal
seiching process.
The surface slope indicates that the sealevel at Cochamo was 0.07 m higher than at the
mouth, and this value can be taken as the amplitude of the surface seiche. According to the
RGM, the pycnocline deviation ($\eta_l$) is related to the surface elevation ($\eta_0$) in the form
$\eta_1 = -(\rho/\Delta\rho)\ \eta_0$, which implies that for a mean surface perturbation of 0.07 m and a
typical $\Delta\rho$ of 15 kg m$^{-3}$, we obtain a mean $\eta_l$ of  -4.8 m. This finding indicates that the water
piles up at the head of the fjord, likely due to the predominant into the fjord winds in the
region (Fig. 3a) and produces a pycnocline deepening of about 5 m (Fig. 2).
At low frequencies (periods > 1 day), the along-fjord currents spectra displayed a marked
peak in energy centered at 3 days. To explore the origin of this variability, we analyzed the
density profiles along the fjord (Fig. 2) and applied two methods, the RGM and CSM. The
internal phase velocities (*c*) obtained from both methods were similar, and ranged between
0.73 m s$^{-1}$ and 0.87 m s$^{-1}$ (taking into account the mode 1 of CSM for comparison). The high
*c* value was obtained during the spring (November 2008), when the upper layer presented the
lowest densities of the seasons, likely due to high discharge (> 1000 m$^3$ s$^{-1}$). Remarkably, the
stratification is linked to the freshwater input despite no major observed changes in *c* (Fig.
6e-h). The high consistency between the CSM (mode 1) modal speeds and the phase speed
obtained by RGM suggest that rotation do not play a significant role on the along-fjord
dynamics of these oscillations (van der Lee and Umlauf, 2011). But cross-fjord, the
dynamics has been nearly geostrophic, especially at the fjord's mouth (Castillo et al., 2012).
For longer periods (> 10 days), there are evidences of baroclinic oscillations clearly observed
on the along-fjord time series (Fig. 3) and in the averaged spectra (Fig. 4). Recently, Ross et
al. (2015), described a similar periodicity on currents of a southern Patagonian fjord of Chile
associated to Baroclinic Annular variability, a regional feature on the air-pressure in the
region. This mechanism of generation for the 10 days oscillations on the Reloncavi fjord
needs to be verified on future studies.



The internal $T_N$ of the entire fjord displayed periods between 2.9 and 3.5 days. These results
suggest that the accumulation of energy observed in the along-fjord currents are due to the
first mode of an internal seiche oscillation in the fjord. This result could be explained by the
presence of a node at the mouth, where the sealevel amplitude is minimum (Fig. 5) but the
currents are maxima (Figs. 3 and 6). This difference was also observed in the projected
currents ($u_{pj}$, $v_{pj}$) supporting the idea of the presence stationary wave along the fjord.
Additionally, the currents were highly coherent and in phase (Fig. 8) as we expected from a
basin-scale seiche wave like. As a way of estimate the contribution of the internal seiche to
the internal mixing the $K_E$ was enhanced during the into the fjord winds (Figs. 3 and 7),
which were periods when the internal seiche band (3 days) was highly coherent along the
fjord (Fig. 8).
The winds exhibited high coherence with the along-fjord currents until the pycnocline
depths, at frequencies centered at 1 and 3 days (see Fig. S1). To study the extent to which the
wind stress perturbs the pycnocline, we used the Wedderburn number, which is given by the
equation $W = (h_1 / L)Ri$ (Thompson and Imberger, 1980; Monismith, 1986), where
$Ri = g'(h_1 / u_*^2)$ represents the bulk Richardson number, an index of the stability of the upper
layer ($h_1$). The frictional velocity ($u_*$) is obtained from the surface wind stress using the
equation $u_*^2 = \tau / \rho_0$, which results in the equation,

$$W = \frac{h_1^2 \Delta \rho g}{L \tau} \tag{6}$$

According to Thompson and Imberger (1980), this value indicates the effect of the wind
stress on local upwelling in a stratified fluid (i.e., perturbing the pycnocline). Under weak $\tau$
conditions ($W \gg 1$), the wind energy is insufficient to tilt the interface. Under strong $\tau$
conditions ($W \ll 1$), however, upwelling conditions dominate, there by tilting the interface,
which produces conditions favorable to forcing of the internal seiche. The critical conditions
($W \sim 1$) indicate the beginning of upwelling (Thompson and Imberger, 1980; Stevens and
Imberger, 1996), although the ideal transition point occurs at $W = 0.5$ (Monismith, 1986). All
of these conditions were observed during the period of August 2008, as it is shown on Fig. 9.





During strong $\tau$ (~0.3 N m$^{-2}$) conditions, $W = 0.27$ produced intense perturbation of the
pycnocline (Fig. 9a). In contrast, during weak $\tau$ (~0.01 N m$^{-2}$) conditions, a value of $W = 8$
indicates that the wind was too weak to perturb the pycnocline, favoring a seiche damping
process (Fig. 9b). Transition conditions occurred when $\tau \sim 0.1$ N m$^{-2}$ and $W = 0.8$, indicating
that the winds were strong enough to perturb the pycnocline and stop the damping process
(Fig. 9c).
## 5.1    Internal seiche damping
The wind stress changed from a state where $\tau$ was strong enough to actively disturb the
pycnocline ($W < 1$) to a period of nearly calm winds ($W > 1$) between the 16 and 24 August
2008 (Fig. 9). During this period, both the along-fjord currents and temperatures tended to
decay, which is clearly evident in the isolines of these properties at the three sites (Fig. 9).
To study the damping process in detail, we selected the time series of the along-fjord
currents at a depth of 3 m at Cochamo during the above period in August to span the period
of forcing, damping and re-enforcing of the internal oscillation.
Typically, any real oscillations undergo damping, which is given by the equation,
$$x(t) = A\, e^{(-k\,t)} \cos(\omega\, t + \phi) \qquad (7)$$
where $t$ is time and $A$ is the initial amplitude, $k$ is the scale, $\omega = 2\pi/T_N$ and $\phi$ is the phase. In
the case studied here, $\phi = 0$, $A = 8$ cms$^{-1}$, and $T_N = 2.5$ days, which was the internal period at
Cochamo (Fig. 4). The best fit occurred when $k = 1/3$ (Fig. 10).
The time for the initial amplitude $A$ to decay to $A \sim 0$ is the damping time ($T_d$). There was a
good fit (Fig. 10) between the observed current and the curve adjusted with the damping
effect. Here, $T_d = 9.1$ days, which is more than 3 times longer than the natural oscillation
($T_N$); more precisely, $T_d = 3.6\ T_N$ at this site. The observed internal oscillations of the currents
were not completely damped because the winds increased from nearly calm ($W > 1$) to
moderate conditions, which disturbed the pycnocline ($W \sim 1$) and induced the intense


oscillations during the spring (Fig. 6). In the spring, the winds displayed a marked diurnal
cycle that remained during the spring and summer (Castillo et al., 2012). This finding
suggests that the internal seiche (mode 1) process is active without damping because it is
forced daily (Fig. 3). Our findings indicated that the internal seiche process is an active
contributor for the mixing in the Reloncavi fjord, the magnitude of this contribution might be
similar as the tidal forcing. The maximum amplitude of the tidal currents on the Reloncavi
fjord is 10 cm s$^{-1}$ (Valle-Levinson et al., 2007; Castillo et al., 2012), taken the eq. 7 to
estimate the maximum contribution of the tide obtain 5 x 10$^{-3}$ m$^2$ s$^{-2}$ which is similar to the
observed $K_E$ at the mouth (Fig. 7). One example of the dissipation of the energy through this
process was observed previous to 19 August 2008 (Fig. 10), on there the maximum currents
were 0.7 m s$^{-1}$ and through eq. 7, we obtain $K_E$= 7 x 10$^{-3}$ m$^2$ s$^{-2}$ great part of this energy
might be dissipated within the Reloncavi fjord on 9 days. Future studies should focus on
evaluating more precisely the available energy for the mixing process within the fjord and
their effects on other water properties such as the salinity, oxygen or nutrients.
**6**      **Conclusions**
The along-fjord seasonal density structure of the Reloncavi fjord showed small changes in
the stratification. The upper layer presents a persistent stratification from the surface to the
pycnocline base, the latter of which has a mean depth of 2 m near the mouth and 6 m near the
head of the fjord.
The along-fjord sealevel signal showed a 1.3 h energetic peak not related with any tidal
harmonics, additionally at this period the sealevel amplitude at the mouth was significantly
higher than the sealevel at the head of the fjord. This pattern was consistent with the presence
of a barotropic seiche on the Reloncavi fjord.
The local winds stress was able to perturb the along-fjord pycnocline and produce internal
seiche oscillations. The period centered on 3 days was consistent with the first baroclinic
oscillation mode. This mode explained 44% of the variability of the 3 days band. The





oscillation was highly coherent along the fjord and with a phase nearly to 0º, consistent with
a standing wave, like an internal siche, within the Reloncavi fjord.
The internal seiche could be high contributor to the internal mixing within the fjord, in fact
the kinetic energy ($K_E$) associated to the internal seiche was similar to the maximum
contribution of the tides in the along-fjord currents. During winter, the internal oscillations
were present a relatively long period of time with nearly calm winds permit the estimation of
the damping time of the internal seiche which was of 9 days, otherwise during the spring
daily winds continuously forced the pycnocline.
**Data availability**
The installation of the moorings for measuring the current, temperature and sealevel in the
region was approved by the Chilean Navy through permit DS711. No specific permits were
required to install the meteorological station because the location is a publicly controlled site.
This study also did not involve any endangerment to species in the region. The authors
indicated that all data are available to download from a COPAS-SUR Austral website
(http://www. reloncavi.udec.cl/, last access 6 June 2016).The discharge data from the rivers
of Chile are available from the Dirección General del Aguas de Chile website
(http://dgasatel.mop.cl/, last access 1 July 2016). Also, all data sets can be requested from the
corresponding author (Manuel I. Castillo).
**Acknowledgements**
The authors thank the students (from Chile and Sweden) and technicians of the Physical
Oceanography group of the Universidad de Concepcion who collaborated in performing the
field measurements. This study is part of the PFB/31 COPAS-Sur Austral program and
Centro de Investigación en Ecosistemas de la Patagonia by FIP2007-21 and Comité
Oceanográfico Nacional CIMAR-CONA C17F 1107. Manuel I. Castillo was supported by
FONDECYT grant no. 3130639 and by CONICYT-PAI no. 791220005.





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





**Figure captions**
**Figure 1:** Study region and location of the measuring stations. Left insert shows the area of
the Reloncavi fjord (A). The location of the Reloncavi sound (B) is also shown. The right
insert shows the study area (close-up view of A) and the positions of all measurements.
Numbers are CTD stations.
**Figure 2:** Seasonal profiles of density and bathymetry of the region. The upper inserts show
the seasonal mean density profiles in each sub-basin of the fjord (a-d). In the insert below
(e.), the along-fjord bathymetry and sub-basin nomenclature are shown. The black line
represents the mean pycnocline depth, and corresponding standard deviations are represented
by the gray shading.
**Figure 3:** a)Along-fjord wind stress, positive up to the fjord; (b) sealevel, (c)Puelo river
discharge, where the straight line represents the long-term mean and contours of along-fjord
currents at (d) Cochamo, (e) Puelo and (f) the mouth. In the filled contours, the blue (red)
colors indicate a net outflow (inflow).
**Figure 4:** Spectra of along-fjord currents (above) at the mouth (a), Puelo (b) and Cochamo
(c). Here black line indicate the averaged spectra for the upper layer (depths · h1) whereas
gray lines showed spectra for currents at depths > h1. (d) sealevel spectra at the mouth (black
line) and at Cochamo (gray). (e) wind stress spectra for their along-fjord (black) and cross-
fjord (gray) components. At the bottom each panel the 95% of confidence intervals for 48, 24
and 12 degrees of freedom are shown.
**Figure 5:** The left insert shows mean density ($\sigma_t$) within the sub-basins. The pannels to the
right of these show the first 3 baroclinic $\psi_n(z)$ modes and modal speeds obtained from the
CSM analysis (normalized). Note that phase velocity is in [m s-1].
**Figure 6.** Band-passed along-fjord currents. Contours of band-passed (70-90 h) along-fjord
currents. Negative (positive) currents in blue (in red) imply an outflow (inflow). Note the
dotted square at the middle of August it is zooming on figure 9.



**Figure 7.** Projected along-fjord currents and kinetic energy ($K_E$). Here presented the 1 to 3
modal projections of the along-fjord band-passed (60-100 h) currents at Cochamo and at the
mouth. At the bottom, present the $K_E$ estimated using the projected components at the mouth
(red) and at the mouth (black).
**Figure 8.** Coherence and phase wavelet spectra. Time series of along fjord wind-stress (a),
and coherence and phase wavelet spectra for the relation mouth/Puelo (b, c) and
Puelo/Cochamo (d, e). In the contours, the thick black line indicates squared coherence $\geq 0.6$,
only the associated phases were present on the phase wavelet. The thick black curve is the
influence cone for the wavelet estimations.
**Figure 9.** Time-series of along-fjord wind stress ($\tau$) and contours of along-fjord currents (V)
and temperatures (T) at Cochamo, Puelo and the mouth. There are three states of wind stress
based on the Wedderburn number ($W$) with (a) strong $W < 1$, (b) weak $W > 1$ and moderate
$W \sim 1$ winds. Note that contours of the current (V) and temperature (T) for a given location
are plotted together.
**Figure 10.** Damping signal in currents. During a period of weak winds ($W>1$) at Cochamo
(16 to 24 August 2008). The band-pass currents at the 3m depth (black line) was compared
with a damping oscillatory curve $x(t) = A\,e^{(-kt)}cos(\omega t + \phi)$ (gray line). The damping time ($T_d$)
was 3.6 times longer than the fundamental internal period ($T_N$).





**Table titles**
**Table 1:** Characteristic of Reloncavi fjord. The name, mean depth (H) and length (L) of each
sub-basin and for the entire fjord are presented.
**Table 2:** Seasonal statistics of the descriptive parameters of the fjord. Here we present the
mean depth of the upper layer ($h_1$), and densities of the upper ($\rho_1$) and deep layers ($\rho_2$). In
addition, the phase and modal velocities ($c$) and theirs periods (T) estimated using the
Reduced Gravity and Continuously Stratified models are shown.





1    **Table 1.**

| Sub-basin | Description | H [m] | L [km] |
|:---:|:---:|:---:|:---:|
| I | mouth–Marimeli | 440 | 14.0 |
| II | Marimeli – Puelo | 250 | 13.0 |
| III | Puelo–Cochamo | 200 | 17.5 |
| IV | Cochamo–head | 82 | 10.5 |
| **Total** | **mouth -head** | **250** | **55** |





**Table 2.**

| Reduced Gravity Model (RGM) | | | | |
|---|---|---|---|---|
| $h_1$ [m] | $\rho_1$ [kg m$^{-3}$] | $\rho_2$ [kg m$^{-3}$] | $c$ [m s$^{-1}$] | $T$ [days] |
| **Winter** 4.60 ± 0.60 | 1009.72± 4.32 | 1024.62 ± 0.74 | 0.76 ± 0.01 | 3.37 ± 0.03 |
| **Spring** 4.79 ± 0.53 | 1007.63± 5.32 | 1024.78 ± 0.62 | 0.87 ± 0.02 | 2.92 ± 0.03 |
| **Summer** 4.68 ± 0.26 | 1008.77± 3.26 | 1024.78 ± 0.63 | 0.83 ± 0.01 | 3.07 ± 0.02 |
| **Autumn** 4.05 ± 0.41 | 1009.90± 3.92 | 1024.95 ± 0.48 | 0.75 ± 0.01 | 3.38 ± 0.03 |

| Continuous Stratified Model (CSM) | | | | | |
|---|---|---|---|---|---|
| $c_1$ [m s$^{-1}$] | $c_2$ [m s$^{-1}$] | $c_3$ [m s$^{-1}$] | $T_1$ [days] | $T_2$ [days] | $T_3$ [days] |
| **Winter** 0.73 ± 0.11 | 1.46 ± 0.21 | 2.18 ± 0.32 | 3.50 ± 0.25 | 1.75 ± 0.13 | 1.17 ± 0.08 |
| **Spring** 0.87 ± 0.10 | 1.73 ± 0.21 | 2.59 ± 0.31 | 2.94 ± 0.18 | 1.47 ± 0.09 | 0.98 ± 0.06 |
| **Summer** 0.84 ± 0.07 | 1.68 ± 0.13 | 2.52 ± 0.20 | 3.03 ± 0.12 | 1.51 ± 0.06 | 1.01 ± 0.04 |
| **Autumn** 0.77 ± 0.08 | 1.54 ± 0.15 | 2.32 ± 0.23 | 3.30 ± 0.16 | 1.65 ± 0.08 | 1.10 ± 0.05 |



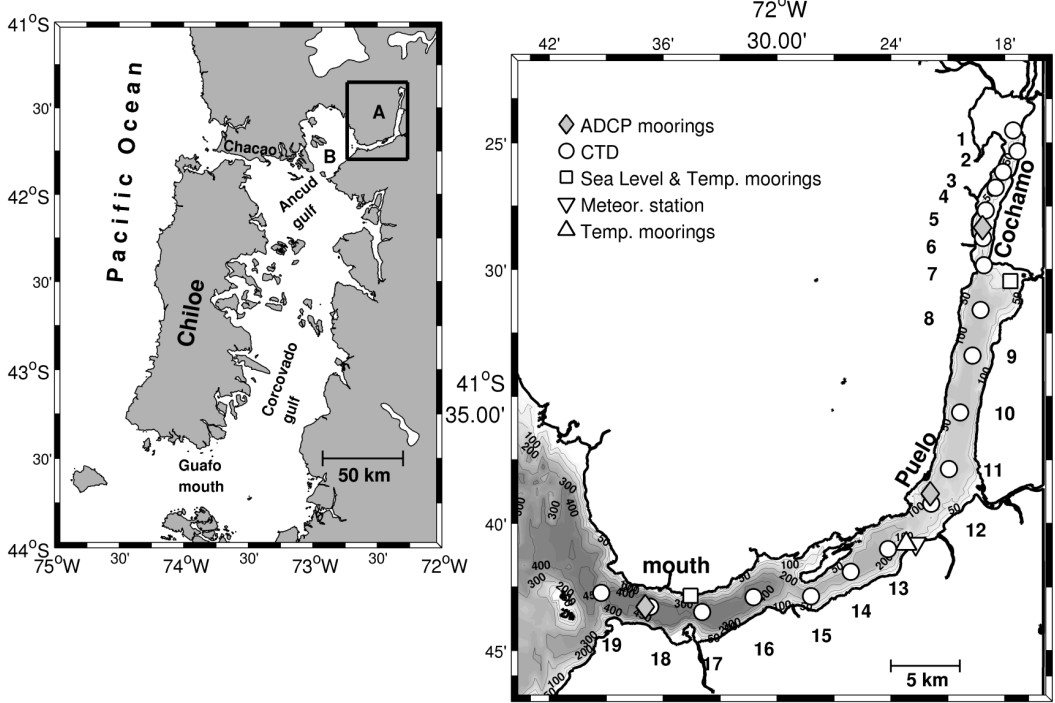

**Figure 1:** Study region and location of the measuring stations. Left insert shows the area of the

Reloncavi fjord (A). The location of the Reloncavi sound (B) is also shown. The right insert

shows the study area (close-up view of A) and the positions of all measurements. Numbers are

CTD stations.



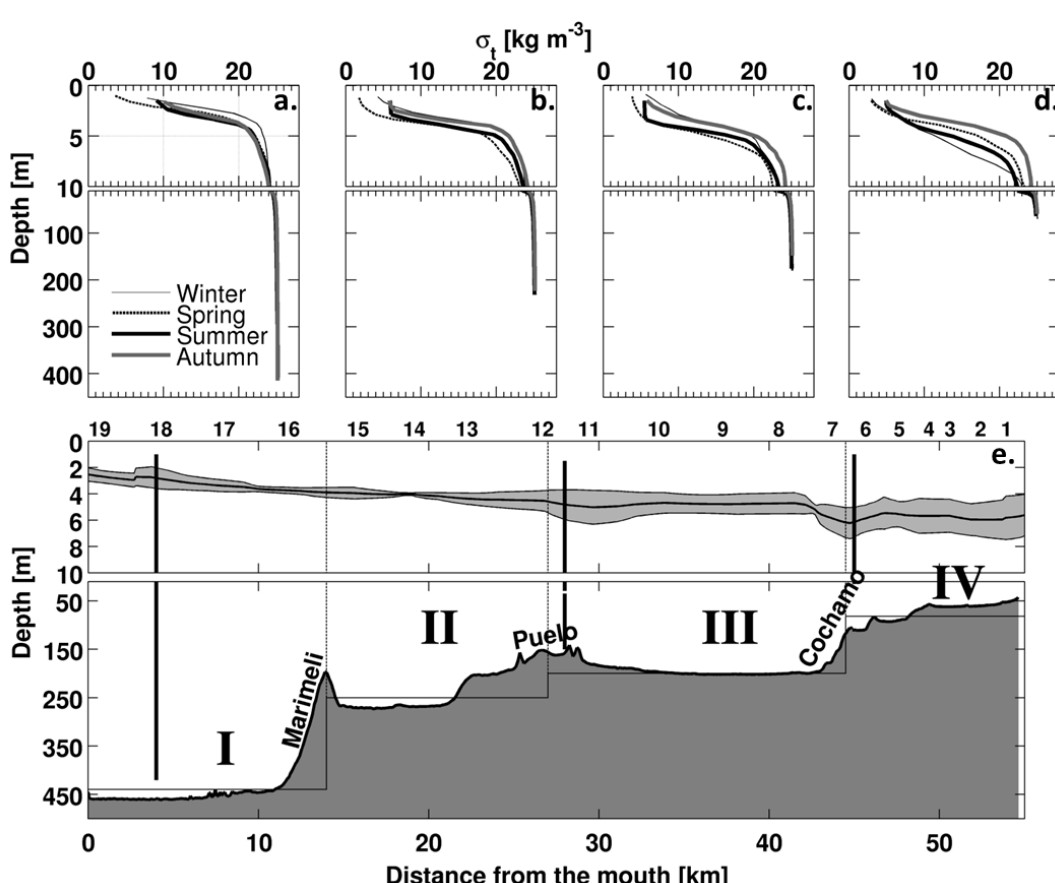

**Figure 2:** Seasonal profiles of density and bathymetry of the region. The upper inserts show the
seasonal mean density profiles in each sub-basin of the fjord (a-d). In the insert below (e.), the
along-fjord bathymetry and sub-basin nomenclature are shown. The black line represents the
mean pycnocline depth, and corresponding standard deviations are represented by the gray
shading.



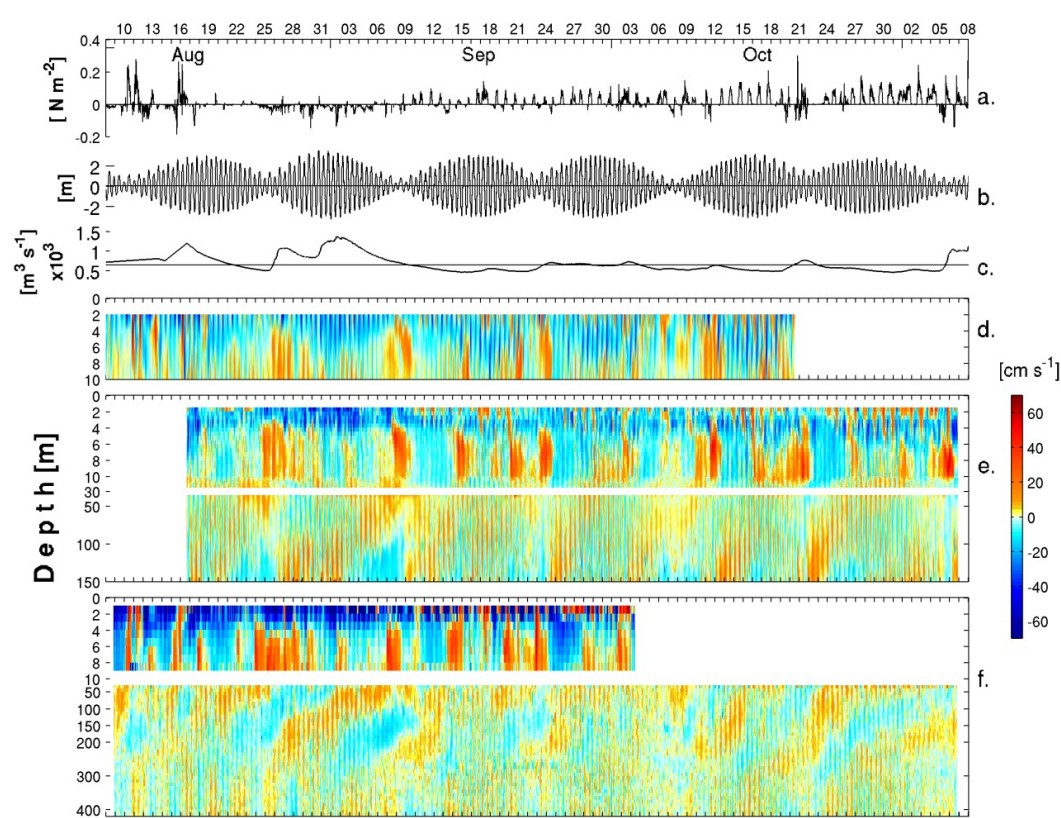

**Figure 3:** a)Along-fjord wind stress, positive up to the fjord; (b) sealevel, (c)Puelo river
discharge, where the straight line represents the long-term mean and contours of along-fjord
currents at (d) Cochamo, (e) Puelo and (f) the mouth. In the filled contours, the blue (red) colors
indicate a net outflow (inflow).



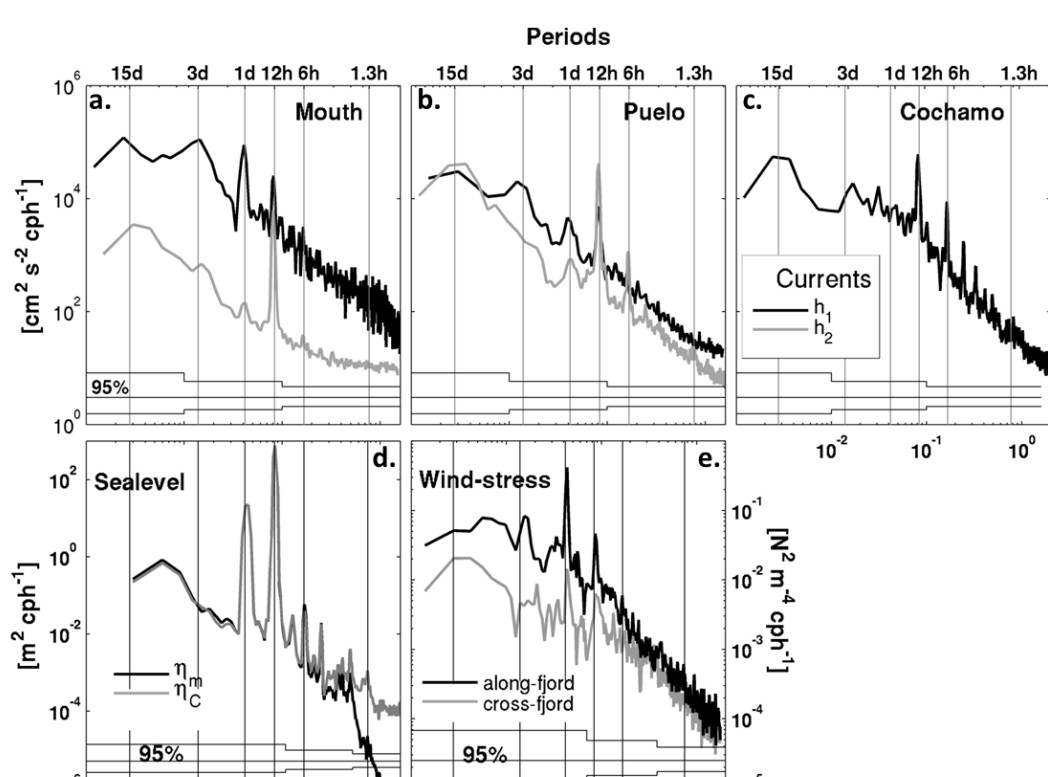

**Figure 4:** Spectra of along-fjord currents (above) at the mouth (a), Puelo (b) and Cochamo (c).

Here black line indicate the averaged spectra for the upper layer (depths ≤ h1) whereas gray lines

showed spectra for currents at depths > h1. (d) sealevel spectra at the mouth (black line) and at

Cochamo (gray). (e) wind stress spectra for their along-fjord (black) and cross-fjord (gray)

components. At the bottom each panel the 95% of confidence intervals for 48, 24 and 12 degrees

of freedom are shown.



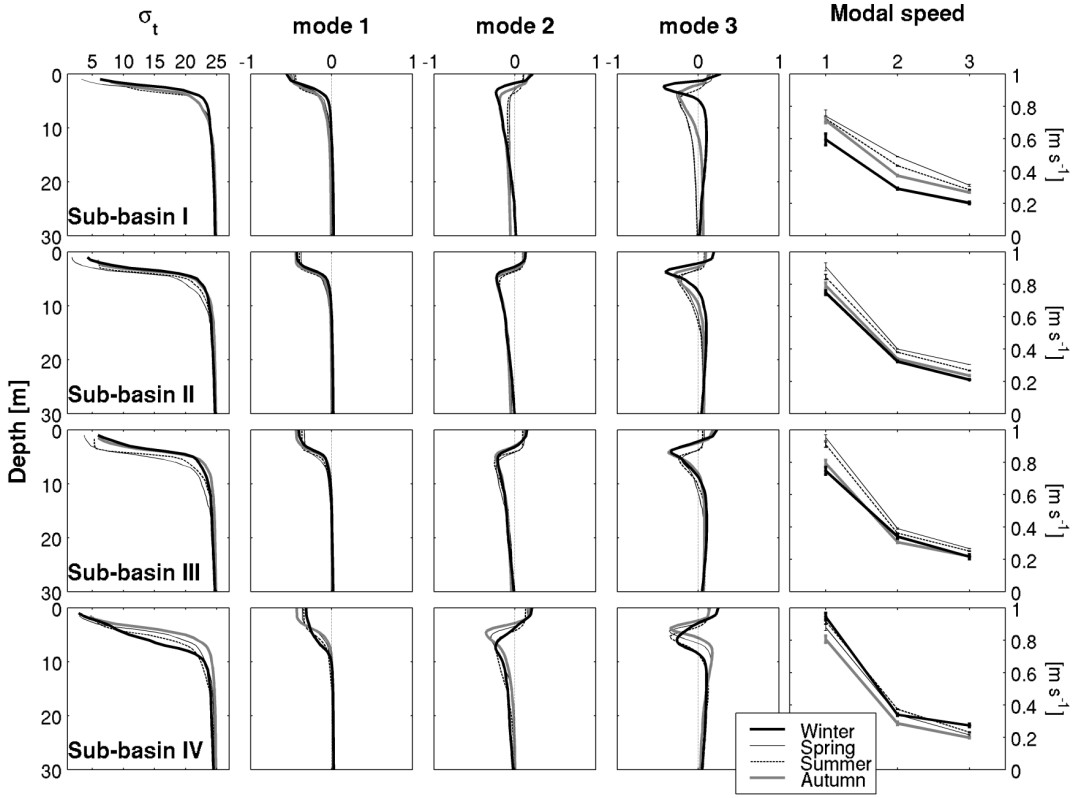

3 **Figure 5:** The left insert shows mean density ($\sigma_t$) within the sub-basins. The panels to the right of

4 these show the first 3 baroclinic $\psi_n(z)$ modes and modal speeds obtained from the CSM analysis

5 (normalized). Note that phase velocity is in [m s-1].





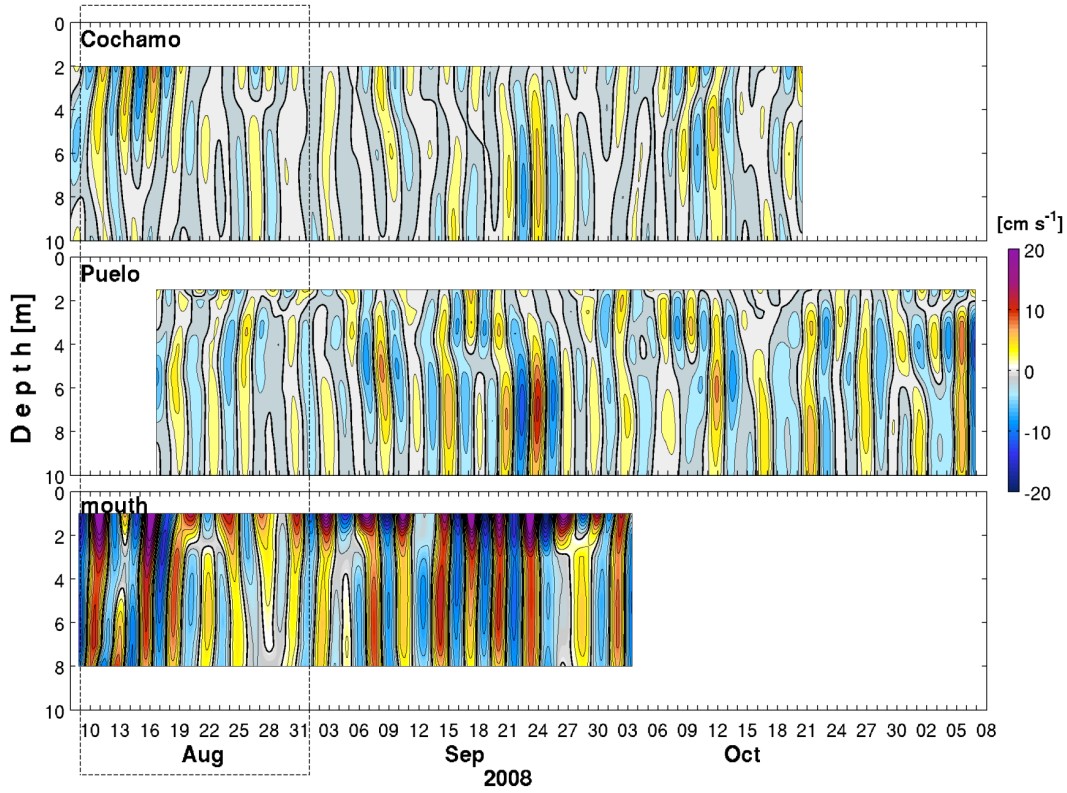

**Figure 6.** Band-passed along-fjord currents. Contours of band-passed (70-90 h) along-fjord currents. Negative (positive) currents in blue (in red) imply an outflow (inflow). Note the dotted square at the middle of August it is zooming on figure 9.



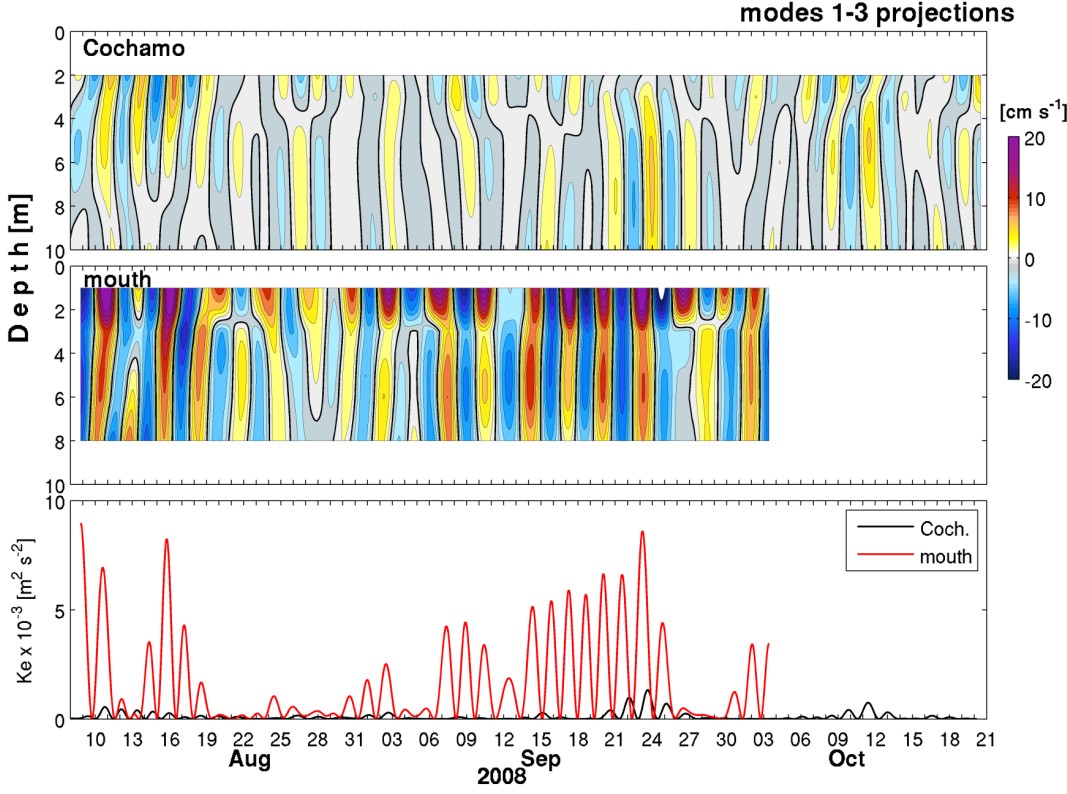

**Figure 7.** Projected along-fjord currents and kinetic energy ($K_E$). Here presented the 1 to 3 modal
projections of the along-fjord band-passed (60-100 h) currents at Cochamo and at the mouth. At
the bottom, present the $K_E$ estimated using the projected components at the mouth (red) and at the
mouth (black).





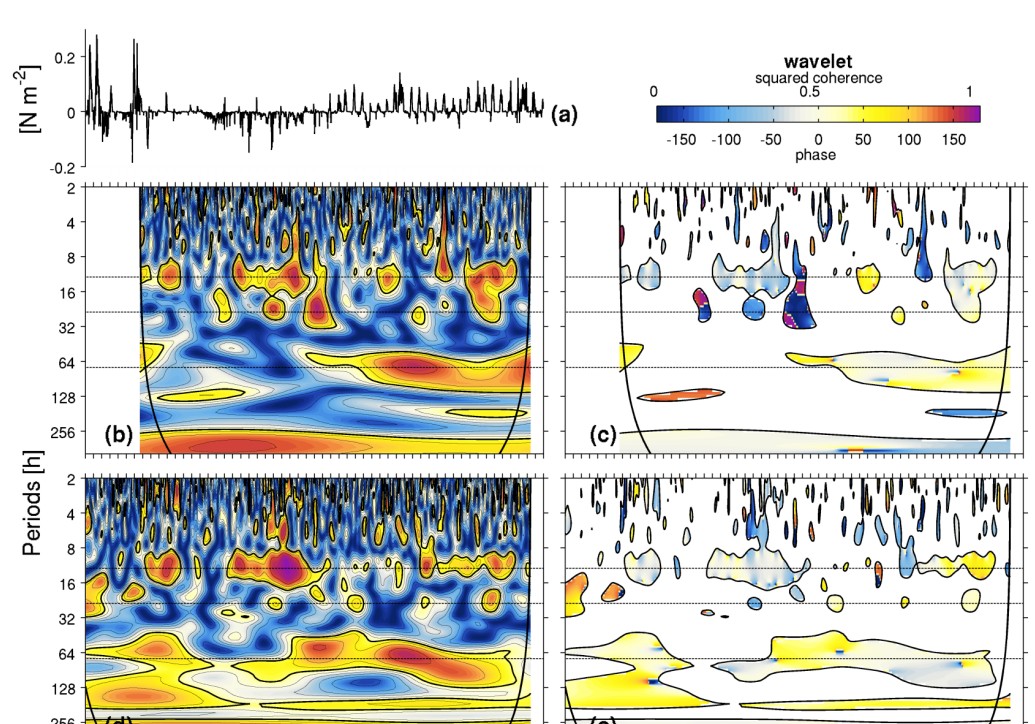

**Figure 8.** Coherence and phase wavelet spectra. Time series of along fjord wind-stress (a), and
coherence and phase wavelet spectra for the relation mouth/Puelo (b, c) and Puelo/Cochamo (d,
e). In the contours, the thick black line indicates squared coherence ≥ 0.6, only the associated
phases were present on the phase wavelet. The thick black curve is the influence cone for the
wavelet estimations.



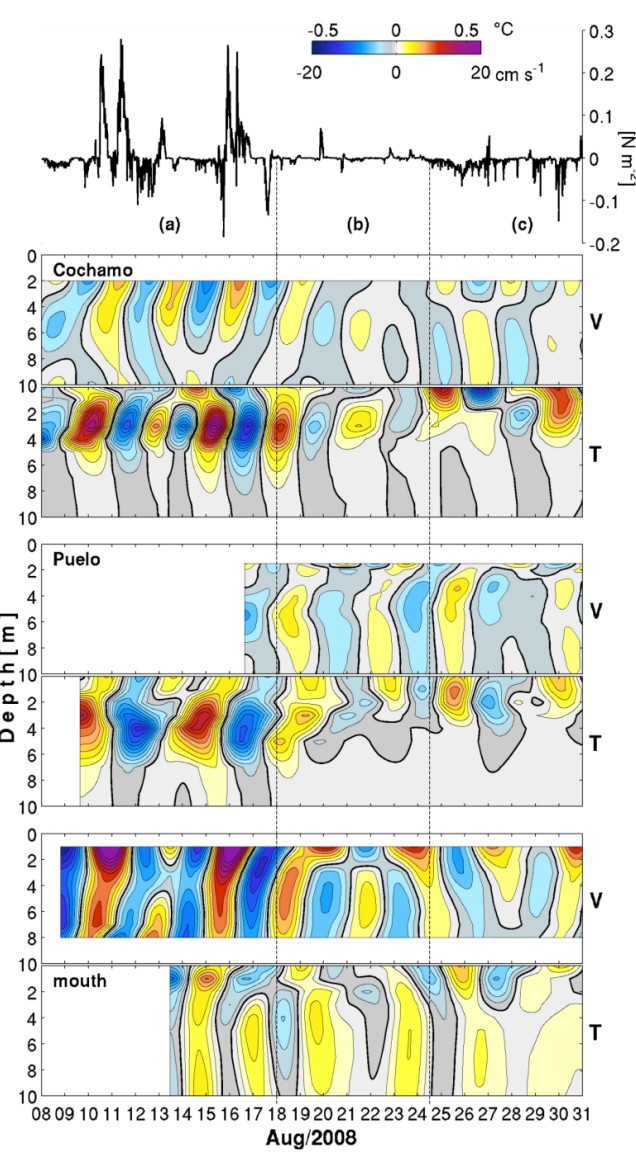

**Figure 9.** Time-series of along-fjord wind stress (τ) and contours of along-fjord currents (V) and
temperatures (T) at Cochamo, Puelo and the mouth. There are three states of wind stress based on
the Wedderburn number (*W*) with (a) strong *W*< 1, (b) weak *W*> 1 and moderate *W*~ 1 winds.
Note that contours of the current (V) and temperature (T) for a given location are plotted
together.





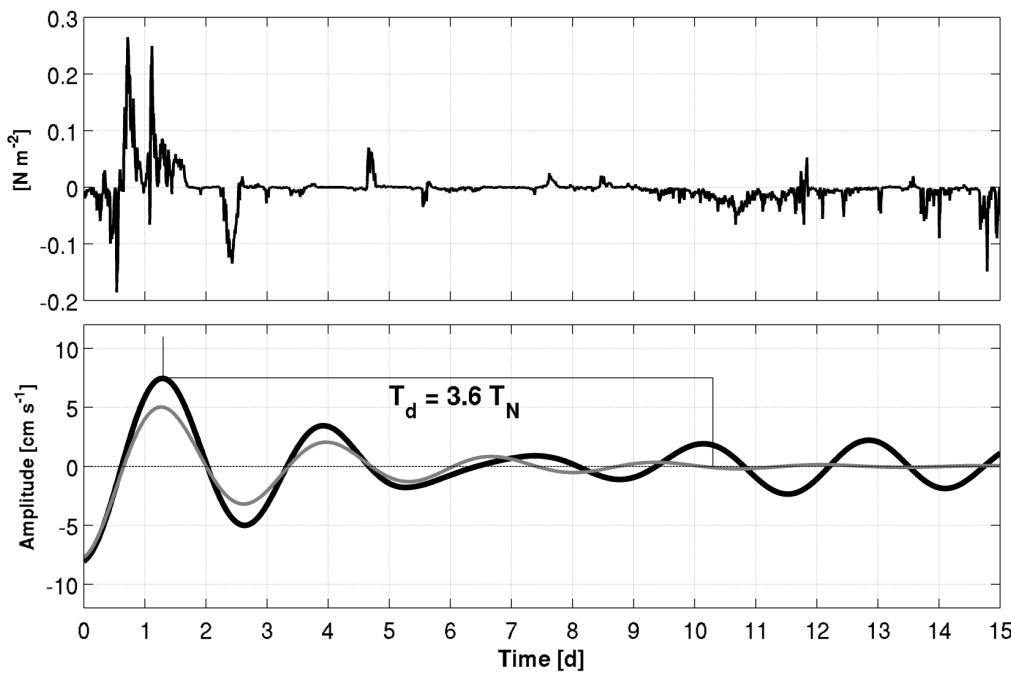

**Figure 10.** Damping signal in currents. During a period of weak winds (*W*>1) at Cochamo (16 to
24 August 2008). The band-pass currents at the 3m depth (black line) was compared with a
damping oscillatory curve $x(t) = A\ e^{(-kt)} cos(\omega t + \phi)$ (gray line). The damping time ($T_d$) was 3.6
times longer than the fundamental internal period ($T_N$).

