# Peer review of "Seiche excitation in a highly stratified fjord of southern Chile: the Reloncaví fjord. Manuel I. Castillo1,2\*, Oscar Pizarro2,3,4 Nadin Ramírez2,4 and Mario Cáceres1 [1]{Escuela de Biología Marina, Facultad de Ciencias del Mar y de Recursos Naturales"

_Ocean Science, 2016_

## Referee Comment (RC1) · Anonymous Referee #1 · 20 Sep 2016

In their manuscript entitled "Seiche excitation in a highly stratified fjord of southern Chile: the Reloncavi fjord", the authors present a detailed observational dataset spanning three months (August-November 2008) to study the variability in the Reloncavi fjord, Chile. Combining in-situ data (ADCP and temperature on three mooring lines, CTD casts) with meteorological and sea level monitoring data, the authors are able to analyze the variability of currents in the fjord. Their analysis demonstrates in particular the presence of internal seiches of period $\sim$ 3 days excited by the wind stress. They are able to infer a damping time of $\sim$ 9 days.

This study is interesting as it provides a thorough analysis of the flow variability in a fjord using detailed observations. The interpretation of the results is based on simple and robust theoretical frameworks, and the conclusions are convincingly drawn. I therefore recommend this manuscript for publication, pending some minor revisions that I list

below.

Minor comments:

1) Lines should be numbered continuously throughout the entire document.

2) Line 9, p4: avoid repetition of the word "forcing"

3) Line 23, p8: You refer to a mean wave speed, but Equ. (1) defines a time. Can you clarify?

4) Line 11, p14: Change the sentence after the parenthesis.

5) Line 1-2, p15: I hardly understand this sentence.

6) Line 24, p15: remove fix this sentence.

7) Line 10-28, p17: Do you really mean Fig. 5, or rather Fig. 4 in this paragraph?

8) Line 1, p18: replace "perturb" with "perturbing"

9) Line 8, p19: "a way of estimating"

10) Line 29 and Fig. 9: it would be very helpful if you would add the time evolution of W in Fig. 9, e.g. in the upper panel superimposed on the wind stress.

11) Line 23, p20: What is the unit of k?

12) Line 6-9, p22: Restate this sentence.
* * *

---

## Referee Comment (RC2) · Anonymous Referee #2 · 6 Oct 2016

General comment

This is an interesting paper which describes the internal seiches in a Chilean fjord on the basis of a data set extending over three months. The analysis follows standard procedures, is competently executed and the results are clearly presented. As far as I can see, there are no arresting, novel results but given the sparsity of observations of internal oscillations in fjords, especially in the extensive fjords of the Chile coast, there would seem to be a fair case for publication in Ocean Science.

I sense that, before publication, there are a number of aspects, detailed below, in which the analysis and presentation of the results could be improved and the interpretation enhanced.

Specific issues

1) My main concern about the paper is that it does not clearly identify the relative importance of the seiche in the overall dynamics and mixing processes in the fjord. There is a strong barotropic tidal forcing which, flowing over the sills, will tend to induce a small (?) M2 internal tide. As the authors indicate, there is also an energetic Estuarine Circulation in response to the considerable freshwater input to the fjord as well as wind-driven motions, notably at the diurnal frequency. In this situation it would be good to know the contribution of the internal seiches in relation to the other components of flow. This might be done by including a plots over time of the kinetic energy in each component (as in fig 7 but for all components).

2) The log-log plots of spectral energy density are not suitable for comparison of the relative variance contribution and could, with advantage, be replaced by "equal variance plots" in which you plot P(f) x f versus log f which do demonstrate the relative magnitude ot the energy in different peaks.

3) Differences in the spectral peaks at ~12h (fig 4) suggest that there is a significant internal tidal response as has been observed in other fjords (e.g.Allen and Simpson, Winant 2010). You could isolate this component either by projecting on to the modes or by cross-spectral analysis of flow in the upper and lower layers.

4) The paper emphasises the consistency of the density structure but it does vary somewhat (~20%) and it would be useful to relate this variation to the changes in stratification due to variations in freshwater input and surface heating/cooling. Presumably salinity is the main control on density but surface heat exchange may also be playing a role ?

5) I was surprised that there is not more evidence of the external seiche which was clearly represented in Gullmar fjord of (Arneborg and Liljebladh 2001) Presumably it is apparent in your results as a weak peak in the sea level spectra which scarcely shows in the velocity data. Is this because your noise level is rather high due to your long sampling interval of 20 minutes which doesn't allow averaging if you want to detect a

78 minute seiche ?

6) The paper is generally well written but the English, which is not always clear and idiomatic, needs some attention. References

Winant C.D. (2010): Two-Layer Tidal Circulation in a Frictional, Rotating Basin, JPO 40(6), 1390-1404.

Allen, G., and J. Simpson, 1998: Reflection of the internal tide in Upper Loch Linnhe, a Scottish fjord. Estuarine Coastal Shelf Sci., 46, 683–701.

---

## Referee Comment (RC3) · Anonymous Referee #3 · 18 Oct 2016

The manuscript "Seiche excitation in a highly stratified fjord of southern Chile: the Reloncavi fjord" describe a project where harmonic oscillations in a semi-enclosed system is analyzed using observational data and an analytical model. I agree with the earlier reviewers. Both that the work is well done and worth publishing, and the suggestions for improvement. To that I would add two more:

1. It would be useful is the principles around fjord seiches is described in more detail, preferable with a conceptual cartoon. It would also be good if the theoretical calculation of expected oscillations were coupled to the the cartoon.

2. Present the resulting harmonic frequencies better and discuss the relative contribution of different sources more clearly. You discuss the effect of tides etc but it's not easy to contrast them to seiches. A table would be preferable.

---

## Author Comment (AC1) · 7 Dec 2016

Here we include the original comment and the answer associated

Reviewer 1: In their manuscript entitled "Seiche excitation in a highly stratified fjord of southern Chile: the Reloncavi fjord", the authors present a detailed observational dataset spanning three months (August-November 2008) to study the variability in the Reloncavi fjord, Chile. Combining in-situ data (ADCP and temperature on three mooring lines, CTD casts) with meteorological and sea level monitoring data, the authors are able to analyze the variability of currents in the fjord. Their analysis demonstrates in particular the presence of internal seiches of period 3 days excited by the wind stress. They are able to infer a damping time of 9 days. This study is interesting as it provides a thorough analysis of the flow variability in a fjord using detailed observations. The

interpretation of the results is based on simple and robust theoretical frameworks, and the conclusions are convincingly drawn. I therefore recommend this manuscript for publication, pending some minor revisions that I list below.

Minor comments: 1) Lines should be numbered continuously throughout the entire document. Answer: We will continuously number the lines of the manuscript (MS) in the new version.

2) Line 9, p4: avoid repetition of the word "forcing" Answer: The word was deleted

3) Line 23, p8: You refer to a mean wave speed, but Equ. (1) defines a time. Can you clarify? Answer: The eq. 1, define the fundamental period of oscillation of the basin which is a better way to estimate the oscillation period of a basin. The phase speed is related with the period based on the linear theory. The scope of the manuscript is based on the period of oscillation of the fjord because appear a marked spectral peak on one characteristic frequency (period) band, the three days. That is based on effective phase speed which take into account the changes of depth and lengths of every sub-basin you must notice that the period is related with the phase velocity (c) in the form showed on the eq. 1. Thus imply that, there is an effective period for the entire basin. We incorporated the terms effective phase speed or phase velocity and effective period on the manuscript to remarks the idea.

4) Line 11, p14: Change the sentence after the parenthesis. Answer: The sentence was changed

5) Line 1-2, p15: I hardly understand this sentence. Answer: Here we describe the percentage of variability explained by the first barotropic mode (known as mode 0), and then for the first three baroclinic modes (modes 1-3). Here the intention is remarks that the nature of the 3 days oscillations is baroclinic. We indicated that the percentage of variance explained in the 3 days band by the mode 0 is only 5% thus the nature of the oscillations is baroclinic.

6) Line 24, p15: remove fix this sentence. Answer: The sentence was removed

7) Line 10-28, p17: Do you really mean Fig. 5, or rather Fig. 4 in this paragraph? Answer: Thanks, we corrected the sentence

8) Line 1, p18: replace "perturb" with "perturbing" Answer: The word was replaced

9) Line 8, p19: "a way of estimating" Answer: The sentence was changed

10) Line 29 and Fig. 9: it would be very helpful if you would add the time evolution of W in Fig. 9, e.g. in the upper panel superimposed on the wind stress. Answer: We will include the time evolution of winds on the Figure.

11) Line 23, p20: What is the unit of k? Answer: The damping coefficient (k) has unit of [s-1] which is consistent with the dimensionless of the exponential and also with the argument of the cosine, here the units of x(t) it is given by the amplitude (A).

12) Line 6-9, p22: Restate this sentence. Answer: The sentence was restated

---

## Author Comment (AC2) · 8 Dec 2016

General comment this is an interesting paper which describes the internal seiches in a Chilean fjord on the basis of a data set extending over three months. The analysis follows standard procedures, is competently executed and the results are clearly presented. As far as I can see, there are no arresting, novel results but given the sparsity of observations of internal oscillations in fjords, especially in the extensive fjords of the Chile coast, there would seem to be a fair case for publication in Ocean Science. I sense that, be for publication, there are a number of aspects, detailed below, in which the analysis and presentation of the results could be improved and the interpretation enhanced.

1) My main concern about the paper is that it does not clearly identify the relative im-

portance of the seiche in the overall dynamics and mixing processes in the fjord. There is a strong barotropic tidal forcing which, flowing over the sills, will tend to induce a small (?) M2 internal tide. As the authors indicate, there is also an energetic Estuarine Circulation in response to the considerable freshwater input to the fjord as well as wind-driven motions, notably at the diurnal frequency. In this situation it would be good to know the contribution of the internal seiches in relation to the other components of flow. This might be done by including a plots over time of the kinetic energy in each component (as in fig 7 but for all components).

Answer The paper was focused on the seiche presence because the subtidal dynamics has been studied by the authors in others works (Castillo et al., 2012, 2016). The estuarine circulation is the main characteristic of the upper circulation (layer between 0 to 30 m) within the fjord. The deep circulation presents the influence of tides on the variability in contrast to the upper layer where the tide represents less than 20% of the variability. Despite the fact that in the first part of the study we shows the total variability of the time series, the scope of the study is center on the 3 day band the limits which were obtained from the two approximations a) the reduced gravity (RGM), and b) continuously stratified (CSM) models.

We will incorporate the contribution in terms of kinetic energy for the total, the tidal, and the 3 days band in order to compare the different parts of the flow, but as we indicate the main variability will be associated to the estuarine flow in the upper layer. Here, the study wants to shows that the internal oscillation of the basin must be include on the estimations of mixing.

Finally, according with the reviewer suggestion we include on the new figure 7, the relative contribution to the kinetic energy of the terms indicated above.

2) The log-log plots of spectral energy density are not suitable for comparison of the relative variance contribution and could, with advantage, be replaced by "equal variance plots" in which you plot P(f) x f versus log f which do demonstrate the relative

magnitude of the energy in different peaks.

Answer Thanks for your help, and as you indicated the equally variance plots helps to identify the main peaks of energy, but less energetic peaks, but notorious, vanishing with this technique (see Figure included).

We think that the log-log spectrum provide us a detailed description of the different kind of oscillations present on the time-series, we are aware about the fact that there are some of those oscillations more energetic than others but even that, the less energetic oscillation like the 1.3h peak in the sealevel is extremely consistent with the presence of a barotropic seiche in the fjord which is not possible to observe using the equal variance spectrum.

3) Differences in the spectral peaks at âĹij12h (fig 4) suggest that there is a sig-niשּׁĄcant internal tidal response as has been observed in other fjords (e.g.Allen and Simpson, Winant 2010). You could isolate this component either by projecting on to the modes or by cross-spectral analysis of flow in the upper and lower layers.

Answer Thanks for the suggestion, in fact the internal tides dynamics is the main topic of the ongoing manuscript from the same authors which take results not only from the Reloncavi fjord also to another fjord of the southern Patagonia. We think that the tidal variability is a different topic and out of the scopes of the manuscript. In the case of the internal tides, recently on the southern Patagonian fjords Ross et al (2014, 2015) showed the relatively importance of the internal tides in the high frequency dynamic of the fjords, indeed in the manuscripts they shows to different ways of forcing for the internal waves: GLOFS and low-frequency changes of barometric pressure. As part of a new project, the group involves in this manuscript is worried about the forcing for one hand tides inducing internal tides due to the pycnocline interaction with the bathymetry for example, and for other hand winds could perturb the pycnocline inducing natural oscillations of the basin. We wants to maintain the manuscript scope on the 3 days oscillation which clearly is due to the natural internal oscillation of the Reloncavi fjord,
this finding is extremely relevant for the region because is the first time that the process it is described for the southern Patagonian fjords.

References:

Ross, L., Pérez-Santos, I., Valle-Levinson, A., and Schneider, W. 2014. Semidiurnal internal tides in a Patagonian fjord. Progress in Oceanography, 129: 19-34.

Ross, L., Valle-Levinson, A., Pérez-Santos, I., Tapia, F. J., and Schneider, W. 2015. Baroclinic annular variability of internal motions in a Patagonian fjord. Journal of Geophysical Research: Oceans, 120: 5668-5685.

4) The paper emphasizes the consistency of the density structure but it does vary somewhat (âĹij20%) and it would be useful to relate this variation to the changes in stratification due to variations in freshwater input and surface heating/cooling. Presumably salinity is the main control on density but surface heat exchange may also be playing a role?

Answer We think that the surface heat exchange may play a role in the upper column, in fact on the brackish water layer, were temperature could be more important to the density instead of salinity. We consider include a description of the relevance of the heating/cooling of the upper layer which has a marked seasonal cycle. You must notice that rivers on the region are colder in winter producing a clear thermal inversion (Castillo et al., 2016) while in summer the surface waters could reach until 18°C probably by heat gained by the solar radiation. But the development of the pycnocline along the seasons is consistent with the freshwater input suggesting that the variability of the density in the upper layer is dominated by the freshwater input instead of the surface heating/cooling.

Reference:

Castillo, M.I., Cifuentes, U., Pizarro, O., Djurfeldt, L., Caceres, M., 2016. Seasonal hydrography and surface outflow in a fjord with a deep sill: the Reloncaví fjord, Chile.

Ocean Sci. 12, 533-544.

5) I was surprised that there is not more evidence of the external seiche which was clearly represented in Gullmar fjord of (Arneborg and Liljebladh 2001) Presumably it is apparent in your results as a weak peak in the sea level spectra which scarcely shows in the velocity data. Is this because your noise level is rather high due to your long sampling interval of 20 minutes which doesn't allow averaging if you want to detect a 78 minute seiche?

Answer As the reviewer indicated, the interval time is too long to properly evaluate the 1.3h barotropic seiche in the fjord but that oscillation was mainly observed on the sealevel spectra because on those instruments the interval was 10 minutes instead of the 20 minutes interval used on currents. In the fjord region of Chile the study of that dynamics has been scarcely studied and the main objective of the study was the 3 days band. Despite that we think that the lo-log spectrum of the sealevel it is a good way to observe insights of the internal seiche thus we decide to maintain the representation of the spectrum in loglog plots.

6) The paper is generally well written but the English, which is not always clear and idiomatic, needs some attention.

Answer After make all the corrections indicated by the reviewers, we will send the manuscript to a native English spoken or we will use the American Journal Experts services ( www.aje.com ) to check the language of the manuscript.

References Winant C.D. (2010): Two-Layer Tidal Circulation in a Frictional, Rotating Basin, JPO 40(6), 1390-1404. Allen, G., and J. Simpson, 1998: Reflection of the internal tide in Upper Loch Linnhe, a Scottish fjord. Estuarine Coastal Shelf Sci., 46, 683–701.

**Fig. 1.** equally variance spectrums of currents, winds and sealevel

---

## Author Comment (AC3) · 8 Dec 2016

Reviewer 3 The manuscript "Seiche excitation in a highly stratiïfied fjord of southern Chile: the Reloncavi fjord" describe a project where harmonic oscillations in a semi-enclosed system is analyzed using observational data and an analytical model. I agree with the earlier reviewers. Both that the work is well done and worth publishing, and the suggestions for improvement. To that I would add two more:

1. It would be useful is the principles around fjord seiches is described in more detail, preferable with a conceptual cartoon. It would also be good if the theoretical calculation of expected oscillations were coupled to the cartoon.

Answer: Thanks for the comment but the seiche (external and internal) are well known mechanism. We will work on a scheme of the dynamics but the manuscripts already

have several figures and we still don't be sure whether or not the scheme will be helpful to understand the dynamics involve.

2. Present the resulting harmonic frequencies better and discuss the relative contribution of different sources more clearly. You discuss the effect of tides etc but it's not easy to contrast them to seiches. A table would be preferable.

Answer: This is related with the observation 1 of the reviewer 2, we expect to show the different contribution of tides and internal seiche using the kinetic energy estimations. Those results will be inserts on the new Figure 7.

---

## Author Response (AR1)

Viña del Mar, Jan 13th of 2017
**REF.** MS No.: os-2016-42

**Dr. Mario Hoppema**

Topic Editor

Ocean Science Discussion

Dear Dr. Hoppemai,

Here we present the revised version of the manuscript "Seiche excitation in a highly stratified fjord of southern Chile: the Reloncaví fjord" (MS). Below you find out, in bold text the Reviewer observations following with our answer to each observation. We include page and line of the changes made on MS but that numbers are according to the Marked manuscript attached after the Answer to the reviewers.

We hope that you find this manuscript is now suitable for publication in Ocean Sciences,

Sincerely,

Dr. Manuel I. Castillo (MIC)

on behalf of myself and my coauthors

**Reviewer 1**

**In their manuscript entitled "Seiche excitation in a highly stratified fjord of southern Chile: the Reloncavi fjord", the authors present a detailed observational dataset spanning three months (August-November 2008) to study the variability in the Reloncavi fjord, Chile. Combining in-situ data (ADCP and temperature on three mooring lines, CTD casts) with meteorological and sea level monitoring data, the authors are able to analyze the variability of currents in the fjord. Their analysis demonstrates in particular the presence of internal seiches of period ~ 3 days excited by the wind stress. They are able to infer a damping time of~9 days. This study is interesting as it provides a thorough analysis of the flow variability in a fjord using detailed observations. The interpretation of the results is based on simple and robust theoretical frameworks, and the conclusions are convincingly drawn. I therefore recommend this manuscript for publication, pending some minor revisions that I list below.**

**Minor comments:**
**1) Lines should be numbered continuously throughout the entire document.**

Answer
We  continuously number the lines of the manuscript (MS).

**2) Line 9, p4: avoid repetition of the word "forcing"**

Answer
The word was deleted on line 93.

**3) Line 23, p8: You refer to a mean wave speed, but Equ. (1) defines a time. Can you clarify?**

Answer
The eq. 1, define the fundamental period of oscillation of the basin which is a better way to estimate the oscillation period of a basin. The phase speed is related with the period based on the linear theory. The scope of the manuscript is based on the period of oscillation of the fjord because appear a marked spectral peak on one characteristic frequency (period) band, the three days. That is based on **effective phase speed**  which take into account the changes of depth and lengths of every sub-basin you must notice that  the period is related with the phase velocity (c) in the form showed on the eq. 1. Thus imply that, there is an effective period for the entire basin. We incorporated the terms effective phase speed in the MS on lines 234-237.

**4) Line 11, p14: Change the sentence after the parenthesis.**

Answer
The sentence was deleted on line 395.

**5) Line 1-2, p15: I hardly understand this sentence.**

Answer
Here we describe the percentage of variability explained by the first barotropic mode (known as mode 0), and then for the first three baroclinic modes (modes 1-3). Here the intention is remarks that the nature of the 3 days oscillations is baroclinic. We indicated that the percentage of variance explained in the 3 days band by the mode 0 is only 5% thus the nature of the oscillations is baroclinic. We finally decide delete the paragraph (line 413) which distract for the main idea of the MS which is the presence and structure of the 3 days oscillation.

**6) Line 24, p15: remove fix this sentence.**
Answer
The sentence was removed

**7) Line 10-28, p17: Do you really mean Fig. 5, or rather Fig. 4 in this paragraph?**
Answer
Thanks, we corrected the sentence on line 549.

**8) Line 1, p18: replace "perturb" with "perturbing"**
Answer
The word was replaced on line 573.

**9) Line 8, p19: "a way of estimating"**
Answer
The sentence was changed on line 611.

**10) Line 29 and Fig. 9: it would be very helpful if you would add the time evolution of W in Fig. 9, e.g. in the upper panel superimposed on the wind stress.**
Answer
We deeply sorry here, because in the on line answer we don't understand that you refers about W or the vertical velocity. Now in the revised form we change the old Fig. 9 and include the time evolution of vertical velocity as arrows on the contours of the along-fjord currents and we made an interpretation of the results on lines 519-523.

**11) Line 23, p20: What is the unit of k?**
Answer
The damping coefficient (k) has unit of $[s^{-1}]$ which is consistent with the dimensionless of the exponential and also with the argument of the cosine, here the units of x(t) it is given by the amplitude (A). We include an explination of that on lines 656-657.

**12) Line 6-9, p22: Restate this sentence.**
Answer
The sentence was deleted

**Reviewer 2**
**General comment this is an interesting paper which describes the internal seiches in a Chilean fjord on the basis of a data set extending over three months. The analysis follows standard procedures, is competently executed and the results are clearly presented. As far as I can see, there are no arresting, novel results but given the sparsity of observations of internal oscillations in fjords, especially in the extensive fjords of the Chile coast, there would seem to be a fair case for publication in Ocean Science. I sense that, be for publication, there are a number of aspects, detailed below, in which the analysis and presentation of the results could be improved and the interpretation enhanced.**

**1) My main concern about the paper is that it does not clearly identify the relative importance of the seiche in the overall dynamics and mixing processes in the fjord. There is a strong barotropic tidal forcing which, flowing over the sills, will tend to induce a small (?) M2 internal tide. As the authors indicate, there is also an energetic Estuarine Circulation in response to the considerable freshwater input to the fjord as well as wind-driven motions, notably at the diurnal frequency. In this situation it would be good to know the contribution of the internal seiches in relation to the other components of flow. This might be done by including a plots over time of the kinetic energy in each component (as in fig 7 but for all components).**

Answer
The paper was focused on the seiche presence because the subtidal dynamics has been studied by the authors in others works (Castillo et al., 2012, 2016). The estuarine circulation is the main characteristic of the upper circulation (layer between 0 to 30 m) within the fjord. The deep circulation presents the influence of tides on the variability in contrast to the upper layer where the tide represents less than 20% of the variability. The peaks of the semi-diurnal is large en evident on the spectrum of the along-fjord currents (Fig. 4) but you must take into account is that the energy is related to the integral of the curve. Although in the first part of the MS

we shows the total variability of the time series. The scope of this study is explain the persistent and notorious accumulation of energy centered at the 3 day band which explained a similar amount of energy than the tidal (semi-diurnal + diurnal) components (see Fig. 7). We compares the kinetic energy ($K_E$) of the tidal, the 3 days band and the modal components of the flow to shows and remarks the relatively importance of the 3 days band which is the band related with the internal seiche oscillation. We believe that the internal oscillation of the basin must be include in future estimations of the mixing or carrying capacity of the fjord. See our new explanation of the contribution of the tidal variability on lines 430-443.

**2) The log-log plots of spectral energy density are not suitable for comparison of the relative variance contribution and could, with advantage, be replaced by "equal variance plots" in which you plot P(f) x f versus log f which do demonstrate the relative magnitude of the energy in different peaks.**

Answer
Thanks for your help, and as you indicated the equally variance plots helps to identify the main peaks of energy, but less energetic peaks, but notorious, vanishing with this technique (see Figure included).

We think that the log-log spectrum provide us a detailed description of the different kind of oscillations present on the time-series, we are aware about the fact that there are some of those oscillations more energetic than others but even that, the less energetic oscillation like the 1.3h peak in the sealevel is extremely consistent with the presence of a barotropic seiche in the fjord which is not possible to observe using the equal variance spectrum.

**3) Differences in the spectral peaks at ~12h (fig 4) suggest that there is a significant internal tidal response as has been observed in other fjords (e.g.Allen and Simpson, Winant 2010). You could isolate this component either by projecting on to the modes or by cross-spectral analysis of flow in the upper and lower layers.**

Answer
Thanks for the suggestion, in fact the internal tides dynamics is the main topic of the ongoing manuscript from the same authors which take results not only from the Reloncavi fjord also to another fjord of the southern Patagonia. We think that the tidal variability is a different topic and out of the scopes of the manuscript. In the case of the internal tides, recently on the southern Patagonian fjords Ross et al (2014, 2015) showed the relatively importance of the internal tides in the high frequency dynamic of the fjords, indeed in the manuscripts they shows to different ways of forcing for the internal waves: GLOFS and low-frequency changes of barometric pressure. As part of a new project, the group involves in this manuscript is worried about the forcing for one hand tides inducing internal tides due to the pycnocline interaction with the bathymetry for example, and for other hand winds could perturb the pycnocline inducing natural oscillations of the basin. We wants to maintain the manuscript scope on the 3 days oscillation which clearly is due to the natural internal oscillation of the Reloncavi fjord, this finding is extremely relevant for the region because is the first time that the process it is described for the southern Patagonian fjords.

Answer:
This is related with the observation 1 of the reviewer 2, we expect to show the different contribution of tides and internal seiche using the kinetic energy estimations. Those results will be inserts on the new Figure 7.

[revised manuscript text omitted]
** 0.77 ± 0.08 | 1.54 ± 0.15 | 2.32 ± 0.23 | 3.30 ± 0.16 | 1.65 ± 0.08 | 1.10 ± 0.05 |

---

## Author Response (AR2)

Viña del Mar, Jan 24[th] of 2017
**REF.** MS No.: os-2016-42

**Dr. Mario Hoppema**

Topic Editor

Ocean Science Discussion

Dear Dr. Hoppema,

Here we present the revised version of the manuscript "***Seiche excitation in a highly stratified fjord of southern Chile: the Reloncaví fjord***" (MS). Below you find out, in red your comments and in green our answer. We include the number of the line were the changes was made which are marked in yellow into the marked manuscript (attached after the Answer to the reviewers).

We hope that you find this manuscript is now suitable for publication in Ocean Sciences,

Sincerely,

Dr. Manuel I. Castillo (MIC)

on behalf of myself and my coauthors

Referee #1
2) Line 9, p4: avoid repetition of the word "forcing"
Your Answer: The word was deleted on line 93.

MH: This was not implemented in the revised manuscript. Please do.
Ans.
The paragraph was changed on line 93.

Referee#1
3) Line 23, p8: You refer to a mean wave speed, but Equ. (1) defines a time.
Can you clarify?
Your Answer:
The eq. 1, define the fundamental period of oscillation of the basin which is a
better way to estimate the oscillation period of a basin. The phase speed is
related with the period based on the linear theory. The scope of the manuscript
is based on the period of oscillation of the fjord because appear a marked
spectral peak on one characteristic frequency (period) band, the three days.
That is based on effective phase speed which take into account the changes of
depth and lengths of every sub-basin you must notice that the period is related
with the phase velocity (c) in the form showed on the eq. 1. Thus imply that,
there is an effective period for the entire basin. We incorporated the terms
effective phase speed in the MS on lines 234-237.

MH: I understand your explanation, but this does not answer the comment by
the referee. The sentence starting in line 231 mentions the internal phase
velocity and equation (1) in line 234, which should show that velocity but gives a
time instead. Please explain and correct.
Ans.
We re-arrenged the eq. 1 (line 233) to be consistent with the entire
paragraph. Now the effective period (T) it is defined on lines 235-236.

3) Differences in the spectral peaks at ~12h (fig 4) suggest that there is a
significant internal tidal response as has been observed in other fjords
(e.g.Allen and Simpson, Winant 2010). You could isolate this component either
by projecting on to the modes or by cross-spectral analysis of flow in the upper
and lower layers.
Your Answer:
Thanks for the suggestion, in fact the internal tides dynamics is the main topic
of the ongoing manuscript from the same authors which take results not only
from the Reloncavi fjord also to another fjord of the southern Patagonia. We
think that the tidal variability is a different topic and out of the scopes of the
manuscript. In the case of the internal tides, recently on the southern
Patagonian fjords Ross et al (2014, 2015) showed the relatively importance of
the internal tides in the high frequency dynamic of the fjords, indeed in the
manuscripts they shows to different ways of forcing for the internal waves:
GLOFS and low-frequency changes of barometric pressure. As part of a new
project, the group involves in this manuscript is worried about the forcing for one
hand tides inducing internal tides due to the pycnocline interaction with the
bathymetry for example, and for other hand winds could perturb the pycnocline inducing natural oscillations of the basin. We wants to maintain the manuscript scope on the 3 days oscillation which clearly is due to the natural internal oscillation of the Reloncavi fjord, this finding is extremely relevant for the region because is the first time that the process it is described for the southern Patagonian fjords.

MH: Your suggestion to tackle this in a different publication is fine. However, it would be good if you could at least mention the internal tide response in the manuscript and point to the future work.

**Ans.**
**We incorporate a paragraph to indicate the importance of the semi-diurnal signal in the region, and in other studies. We also indicate that it is necessary develop internal-tides studies on the entire Patagonian region on lines 482-491.**

4) The paper emphasizes the consistency of the density structure but it does vary somewhat (~20%) and it would be useful to relate this variation to the changes in stratification due to variations in freshwater input and surface heating/cooling. Presumably salinity is the main control on density but surface heat exchange may also be playing a role?
Your answer:
We think that the surface heat exchange may play a role in the upper column, in fact on the brackish water layer, were temperature could be more important to the density instead of salinity. We consider include a description of the relevance of the heating/cooling of the upper layer which has a marked seasonal cycle. You must notice that rivers on the region are colder in winter producing a clear thermal inversion (Castillo et al., 2016) while in summer the surface waters could reach until 18ºC probably by heat gained by the solar radiation. But the development of the pycnocline along the seasons is consistent with the freshwater input suggesting that the variability of the density in the upper layer is dominated by the freshwater input instead of the surface heating/cooling.

MH: It is not clear to me whether you actually made any changes to the manuscript addressing this issue. Please explain and show where in the manuscript this has been added.
**Ans.**
**We incorporate a similar explanation that above in the manuscript on lines 446-452.**

1. It would be useful is the principles around fjord seiches is described in more detail, preferable with a conceptual cartoon. It would also be good if the theoretical calculation of expected oscillations were coupled to the cartoon.

Your Answer:
Thanks for the comment but the seiche (external and internal) are well known mechanism. We will work on a scheme of the dynamics but the manuscripts already have several figures and we still don't be sure whether or not the scheme will be helpful to understand the dynamics involve.

MH: I agree with the referee that it would be useful to provide a short description of the theory of seiches whether or not with cartoon.
**Ans.**
**We add a brief but concise explanation of the seiche basic dynamics on lines 497-506.**

Below are my comments after editor review:
Please use following format for references in the text: (Watson, 1904) (Smith et al., 2004) (comma after name or after et al.)
**Ans.**
**The format of the references in the text was checked and corrected accordingly with the examples along the entire document, but the changes were not marked because were too much marks.**

In the Methods section 3.1 please provide precisions of the different measurements.
**Ans.**
**The precision of the measurements is already included on supporting Table S2, we indicate that on lines 151.**

L24-25 I suggest: We describe a seiche process based on current, temperature and sea level data obtained from the Reloncavi fjord (41.6° S, 72.5° W) in southern Chile.
**Ans.**
**The sentence was changed on lines 24-26.**

L26 define ADCP here (used for the first time)
**Ans.**
**Now we define Acoustic Doppler Currentmeter Profiler (ADCP) on L26.**

L53 For better understanding I think this should be changed to: It is possible to find resonant basin modes, but only …
**Ans.**
**The sentence was changed on L53-L54.**

L61 Shouldn't this be: "… IN the pycnocline"
**Ans.**
**We sentence was changed on L61**

L64 amplitude
**Ans.**
**We sentence was changed on L64**

L97-99 To avoid questions at this point, change to: "The objective of this study was to address how these oscillations affect the temporal and spatial dynamics of currents and temperature, and how these oscillations are forced."
**Ans.**
**We sentence was changed on L97-L98**

L103 the northernmost fjord
**Ans.**
**Changed L102**

L137 Current (instead of: Currentmeter)?
**Ans.**
**We delete the sentence because ADCP was defined previously**

L140 upward looking (instead of: upper-looking)?
**Ans.**
**Changed L139**

L184 stabilized (not: stabilization)
**Ans.**
**Changed L182**

L202-204 "To focus the study on these perturbations, we used a cosine-Lanczos band-pass filter with half amplitudes at 60 h and 100 h (see results for the justification of the selected band)." You do not just use a filter. With what did you use this filter? I think with the time series data. Please also mention this in the sentence.
**Ans.**
**We indicate in a way explicit that the time series of currents an temperature were band-passed on L200-L201.**

L204-205 "As part of the results, the band-passed time series of the current and temperature data are shown (COPAS-SUR Austral, 2012)." I do not understand this. Where are they shown? Or do you mean something different?
**Ans.**
**Thanks for the comment that was a mistake, now we refer to Fig. 6 and Fig. 9, which showed the band-passed time series change was made on L203.**

L228 "… for each sub-basin, and season" or "… for all sub-basins, and seasons"
**Ans.**
**Changed L227**

L273 … generally directed out of the fjord … (insert: directed)?
**Ans.**
**Changed L273**

L369-371 "In addition, the density structure showed a condition of continuous stratification in the upper layer along the seasons (Fig. 5)." This is repeated info. Exactly the same is written at the beginning of this paragraph. Please delete one or the other.
**Ans.**
**The entire sentence was deleted on L369.**

L378 delete comma
**Ans.**
**Deleted.**

L379-380 If I understand correctly, change to: "Mode 1 was highly baroclinic, changing sign at nearly 10 m (sub-basin I) and 15 m (sub-basin IV)." (Please note that this is not easily visible at Fig.5)
**Ans.**
**Changed on L377**

L381-382 This was strange. Please change to: "For depths > 30 m (not shown in Fig. 5) the internal modes do not show significant variability."
**Ans.**
**Changed on L380**

L387 Change to: Like the internal speeds (c) …
**Ans.**
**Changed on L386**

L399-401 I suggest to slightly change this to: "The vertical patterns at the three locations show inflow-outflow intermittence along the whole time series; also most of these along-fjord structures seem to develop an inclination which indicates the baroclinic nature of this pattern."
**Ans.**
**The suggested changes were implemented on L398-L400**

L404 depth
**Ans.**
**Changed**

L405 decide (instead of: confirm)
**Ans.**
**Changed**

L409-410 I am not sure what you mean with Figure 7. I think you mean to show the agreement between the currents, correct? Then please change this sentence to: "The agreement between the 3 days band-pass and the projected along-fjord currents at the mouth is shown in Fig. 7."
**Ans.**
**Thanks for the suggested change which was insert on L408-L409.**

L410-412 "Using only the first three modes, it was possible to explain more than 70% of the band-pass variability; changes in the outflow and inflow were highly consistent and the intensifications at the surface were clearly shown by the projected modes."
**Ans.**
**Thanks for the suggested change which was insert on L409-411.**

L418-420 Please modify this sentence to: "The vertically averaged KE obtained with 3 days band-pass was higher than that generated with the other the components (modes 1-3); the maximum was observed in the period 9-18 August (Fig. 7), which is consistent with the wind-stress intensification shown in Fig. 3a."
**Ans.**
**Thanks for the suggested change which was insert on L417-419.**

L420-422 "During that period, the modal KE was about one third of the 3 days band-pass kinetic energy; this ratio was higher (i.e. ca. 50%) during September."
**Ans.**
**Thanks for the suggested change which was insert on L420-421.**

L423 delete: relatively (I think this is unnecessary here)
**Ans.**
**Changed**

L423 … was estimated by summing up the …
**Ans.**
**Changed on L422**

L428-429 To describe the temporal variability of this high coherence, …
**Ans.**
**Changed on L426**

L431 (only referred to as coherence hereafter)
**Ans.**
**Changed on  L429**

L431 spectra
**Ans.**
**Changed on L429**

L435 It is not clear to me what opposite winds are here. Please clarify.
**Ans.**
**We refer to the period with down-fjord (to the mouth) winds, we delete "opposite" and explain that on L433.**

L444-445 up-fjord wind
L446 down-fjord winds
**Ans.**
**Both were Changed on L442-444**

L449 delete: here
**Ans.**
**The entire paragraph was changed to incorporate the observation about the heat/cooling effect on the surface column density on L446-452.**

L453-454 I do not understand this sentence: "This change reached depths consistent with the pycnocline (Fig. 2)" Do you mean: "This change reached depths down to the pycnocline (Fig. 2)"
**Ans.**
**Thanks for the suggested change which was insert on L456.**

L463 implies
**Ans.**
**Changed on L465**

L464 velocity patterns
**Ans.**
**Changed on L466**

L469 Change to: We used data collected in one of the most extensive studies ever conducted …
**Ans.**
**Changed on L471**

L476 major (instead of: key)
**Ans.**
**Changed on L478**

L487 interaction
**Ans.**
**Changed**

L488 (not a peak)
**Ans.**
**Changed on L510**

L608 … (Fig. 10); then the maximum
**Ans.**
**Changed on L630**

L609 KE= 7 x 10-3 m2 s-2, meaning that a great part of this energy might …
**Ans.**
**Changed on L631**

L610-612 This sentence belongs to the Conclusions section.
**Ans.**
**Changed on L659-661**

L617 shows (instead of: presents)
**Ans.**
**Changed on L637**

L626 local wind stress
**Ans.**
**Changed on L646**

L629 with a phase close to 0º, …
**Ans.**
**Changed on L649**

L630 seiche (typo)
**Ans.**
**Changed on L650**

L632 strong (instead of: high)
**Ans.**
**Changed on L652**

L635 … winds, which permitted the estimation …
**Ans.**
**Changed on L655**

L636 … seiche being 9 days; otherwise …
**Ans.**
**Changed on L656**

L680 J. Geophys. Res.
L686-687 Prog. Oceanogr.
L689 Change the journal name to: Deep-Sea Research Part A
L702 Prog. Oceanogr.
L751 Prog. Oceanogr.
L754 Prog. Oceanogr.
**Ans.**
**All the upper issues on the References section was changed, and additionally we checked along the entire section**

L797, 799, 802, 803, 820 panel, panels (instead of: inserts)
**Ans.**
**Changed on all the captions**

L808-811 I suggest to change the caption slightly to: a) Along-fjord wind stress, positive up to the fjord, (b) sea level, (c) Puelo river discharge, where the straight line represents the long-term mean. Contours of along-fjord currents at (d) Cochamo, (e) Puelo and (f) the mouth; in the filled contours, the blue (red) colors indicate a net outflow (inflow).
**Ans.**
**Changed on L833-836**

L813-818 Please change according to: Spectra of along-fjord currents (top) at (a) the mouth, (b) Puelo and (c) Cochamo. Here the black lines indicate the averaged spectra for the upper layer (depths ≤ h1) whereas the gray lines show spectra for currents at depths > h1. (d) sea level spectra at the mouth (black line) and at Cochamo (gray). (e) wind stress spectra for their along-fjord (black) and cross-fjord (gray) components. At the bottom of each panel the 95% of confidence intervals for 48, 24 817 and 12 degrees of freedom are shown.

**Ans.**
**Changed on L838-843**

Figure 7. Please add a, b and c to the panels and use these in the caption. The caption is not clear about what is actually shown. Please describe in the caption exactly what we see in the figure.
**Ans.**
**We completely change the caption to explain each panel showed on the figure on L853-856.**

L833-837 (Caption of Fig. 8) Please modify to include:
Coherence and phase wavelet spectra. (a) Time series of along-fjord wind stress, and (b, c, d, e) coherence and phase wavelet spectra, for the relation mouth-Puelo (b, c) and Puelo-Cochamo (d, e). In the contours, the thick black line indicates squared coherence $\geq 0.6$, only the associated phases were present on the phase wavelet. The thick black curve is the influence cone for the wavelet estimations.
**Ans.**
**Changed on L858-862.**

L841 … and (c) moderate W ~1
**Ans.**
**Changed on L866**

L846 Change to: Damping signal in currents during a period of weak …
**Ans.**
**Changed on L871**

L847 … at 3m depth …
**Ans.**
**Changed on L872**

L859 Characteristics
**Ans.**
**Changed on L884**

[revised manuscript text omitted]
** | 0.77 ± 0.08 | 1.54 ± 0.15 | 2.32 ± 0.23 | 3.30 ± 0.16 | 1.65 ± 0.08 | 1.10 ± 0.05 |

---

## Author Response (AR3)

Viña del Mar, Jan 27[th] of 2017
**REF.** MS No.: os-2016-42

**Dr. Mario Hoppema**

Topic Editor

Ocean Science Discussion

Dear Dr. Hoppema,

Here we present the revised version of the manuscript "***Seiche excitation in a highly stratified fjord of southern Chile: the Reloncaví fjord***" (MS). Below you find out, your comments and in red our answers. We include the number of the line were the changes was made which are marked in yellow into the marked manuscript (attached after the Answer to the reviewers).

We hope that you find this manuscript is now suitable for publication in Ocean Sciences,

Sincerely,

Dr. Manuel I. Castillo (MIC)

on behalf of myself and my coauthors

L380 I think (Fig. 5) at the end of the sentence should be deleted, because it is already mentioned earlier and particularly, that it is not shown.

Ans.

The sentence was deleted on L380.

L446-447 … may seasonally play a role in the upper column …

Ans.

Thanks for the comment, the phrase was changed on L446-447.

L447 in the region

Ans.

Changed on L446

L484 other fjord regions

Ans.

Changed on L484

L487 Recently, Ross et al. (2014; 2015) showed …

Ans.

Changed on L487

L488-491 I suggest to change this sentence to: The importance of the internal tides on the southern Patagonian fjords is unknown and future research should be conducted to determine its contribution to the dynamics of currents and mixing.

Ans.

Thanks for you suggestion, the paragraph was changed and highly contributed to improve the main idea of the paragraph on L488-490.

L497-499 The basic dynamics of a barotropic seiche in a fjord originate from winds tilting the along-fjord surface and piling up water at the head of the fjord. The entire fjord basin begins to oscillate after the cessation of the wind.

Ans.

The phrase was changed on L496-498.

L500-502 I do not understand this sentence. I am not sure whether my correction conveys what you meant to write. Please correct again if necessary : During a baroclinic seiche, winds events perturb the pycnocline to induce oscillations with a period commensurate with the fjord stratification (Djurfeldt, 1987).

Ans.

I was agrre with your suggestion, and change the sentence accordingly on L500-501.

L646 Local wind stress …

Ans.

Changed on L645

L844 … 95% confidence intervals … (delete: of)
Ans.
Changed on L843.

L845 for 48, 24 and 12 degrees of freedom are shown. (delete: 817)
Ans.
Oh, thanks that was a mistake. The number was deleted on L844.

Thanks and best wishes
Mario Hoppema

[revised manuscript text omitted]